# Vibronic coherence contributes to photocurrent generation in organic semiconductor heterojunction diodes

Qingzhen Bian [1]*, Fei Ma [2], Shula Chen[1], Qi Wei [3], Xiaojun Su[2], Irina A. Buyanova [1], Weimin M. Chen [1], Carlito S. Ponseca Jr[1], Mathieu Linares[4], Khadga J. Karki[2], Arkady Yartsev[2] & Olle Inganäs[1]*

Charge separation dynamics after the absorption of a photon is a fundamental process relevant both for photosynthetic reaction centers and artificial solar conversion devices. It has been proposed that quantum coherence plays a role in the formation of charge carriers in organic photovoltaics, but experimental proofs have been lacking. Here we report experimental evidence of coherence in the charge separation process in organic donor/acceptor heterojunctions, in the form of low frequency oscillatory signature in the kinetics of the transient absorption and nonlinear two-dimensional photocurrent spectroscopy. The coherence plays a decisive role in the initial ~200 femtoseconds as we observe distinct experimental signatures of coherent photocurrent generation. This coherent process breaks the energy barrier limitation for charge formation, thus competing with excitation energy transfer. The physics may inspire the design of new photovoltaic materials with high device performance, which explore the quantum effects in the next-generation optoelectronic applications.

[1] Department of Physics, Chemistry and Biology (IFM), Linköping University, 58183 Linköping, Sweden. [2] Division of Chemical Physics, Lund University, 22100 Lund, Sweden. [3] Institute of Applied Physics and Materials Engineering, University of Macau, Macau SAR, China. [4] Department of Theoretical Chemistry and Biology, KTH Royal Institute of Technology, 10691 Stockholm, Sweden. *email: qingzhen.bian@liu.se; olle.inganas@liu.se

In organic photovoltaics (OPVs), where the active layer consists of a nanostructured blend of donor (D) and acceptor (A) semiconductors, the key photophysical process is the generation of free charge carriers. Despite extensive studies indicating that the charge transfer (CT) state determines the photovoltage of OPVs, there are reports on the absence of this state in some efficient OPV material systems, claiming that long-range charge separation may occur directly from excitons generated within the polymer domain[1,2]. Intra-molecular or inter-molecular delocalization of charge may result in a decreased Coulomb binding[3,4], allowing excitons (EX) to access more delocalized charge separated (CS) states, which are postulated to be the main precursors of free charge carriers[5,6]. Time-resolved experiments have found evidence for an ultrafast mobile charge generation on a sub-100 fs time scale[7,8].

Based on recent results, it has been proposed that coherent vibrational motion of the donor and/or fullerene may contribute to the primary step of the photoinduced charge-separation process[9,10]. Other studies demonstrate measurement results that suggest coherences are too short lived to contribute significantly[11]. Nevertheless, the presence of other ultrafast processes (coherent intrachain energy transfer etc.) can contribute to the charge separation in the nanoscale systems[12–14]. However, a definitive correlation of oscillatory signals at femtosecond timescale to the charge separation and photocurrent generation is still lacking. Furthermore, the fundamental question about the structural factors that could be used to enhance a coherence contribution in photocurrent generation processes remains unanswered.

Here, we have studied ternary organic photovoltaic blends, with two polymer donors (**D1**, **D2**) and one fullerene (**A1**) or non-fullerene acceptor. Significantly enhanced photocurrent and fill factor of ternary devices have been observed, which cannot be explained by optical absorption and electronic transport properties. We have used transient spectroscopy techniques to unravel the materials' electronic structure, dynamics, and their contribution to the functionality of the devices. Insights on the timescale and processes leading to the formation of mobile charge carriers can be obtained using ultrafast transient absorption spectroscopy (TA) with sub-50 femtosecond (fs) temporal resolution, complemented by transient photoluminescence, transient photoconductivity and transient two-dimensional photocurrent spectroscopy (2DPS). Both films and working devices have been investigated to confirm the consistency of the transient behavior. We have found an oscillatory transient feature responsible for the ultrafast charge separation after the primary excitation, and the frequency of oscillation encompasses a number of low frequency vibrational modes of one of the donor units. Two of the modes at frequencies 360 and 200 cm$^{-1}$ features prominently in the transients indicating their importance in the ultrafast charge separation after the primary excitations. Our 2D photocurrent results clearly demonstrates a distinctive coherent photocurrent generation that persists for the initial ~200 fs in a high performance device, indicating that the vibronic (electronic-vibrational) coherences directly contributes to the formation of charge carriers and enhancement in the photocurrent in the device.

## Results

**Ultrafast charge separation**. We focus on two optimized ternary blends (**Ternary (L)**, D1:D2(L):A1 = 9:1:10, D2 with low molecular weight; **Ternary (H)**, D1:D2(H):A1 = 7:3:10, D2 with high molecular weight) and two reference binary blends (**D1:A1**, **D2:A1**) (Supplementary Fig. 1). Figure 1a shows their molecular structures and Fig. 1b is the steady and transient state absorption spectra of materials. The energy transfer between two donors in the ternary blend demonstrates enhanced excited state population (Supplementary Fig. 2). Furthermore, high molecular weight increases planar conformation and polaron delocalization[15]. Both of these advantages increase the possibility to explore the primary excited states dynamics (Fig. 1c).

We have used near-infrared probe pulses to study the spectral evolution of transient species (Supplementary Fig. 3). In this spectral range, the TA signal consists of the excited state absorption (ESA), stimulated emission (SE), and photo-induced absorption (PIA) of the charged species. Specifically, the PIA band of excited singlet (1500 nm) and charge carriers (1140 nm) can be clarified in the extended infrared region (Supplementary Fig. 4). The infrared kinetics (Fig. 2a) demonstrate that the 860 nm PIA band, increases simultaneously with the singlet exciton dissociation in the initial picoseconds (ps), indicating that the ultrafast charge transfer occurs between the singlet exciton states and the 860 nm PIA band species. Thus, we focus on the 860 nm signal, whose kinetics predominantly reflects charge transfer excitons, and provide a clear measurement of charge carrier generation from initially photogenerated excitons on the polymer. To follow the ultrafast charge dynamics, single-wavelength kinetic measurements have been employed with improved time-resolution and signal-to-noise level, using excitation and probe wavelengths at 550 and 860 nm, respectively. For the blends (Fig. 2b), the kinetic curve at early times is dominated by ESA and later converts into the PIA of charged species. In the reference binary blends, the PIA signal reaches the maximum at ~25 ps for D2:A1 and ~10 ps for D1:A1. Furthermore, the PIA signal in D1:A1 is much stronger than that in D2:A1. In both ternary blends, the PIA signal reaches a similar high-amplitude maximum at ~10 ps, indicating that the PIA signal in the ternary blends is mainly due to the positively charged species on the D1.

In contrast to the binary blends, the kinetics of the ternary blends exhibit a fast decay (~50 fs) of the initially-formed ESA. Furthermore, this component is much more pronounced in the Ternary (H) blend. This short timescale (~100 fs) is consistent with the recently resolved ultrafast charge motion (~150 fs) at the donor/acceptor interface[16], and it is attributed to rapid quenching of the excited states due to an ultrafast charge separation (CS) process (Fig. 1c). Particularly, this ultrafast charge separation is nearly temperature independent (Supplementary Fig. 5; Supplementary Table 1). It has been proposed that hybridization of donor and acceptor states can reduce the free energy barrier, making the excited state dissociate in an adiabatic process[17]. Such ultrafast charge separation reduces the Coulomb interaction between the electron-hole pairs and helps in the generation of free charges. For the Ternary (H) blend (Fig. 2a), there is an ultrafast charge carrier (1140 nm) rise kinetics in the initial 0.2 ps. Furthermore, the difference in the kinetics between the singlet and charge carrier in the initial rise can be traced after 100 fs. This timescale is consistent with the resolved charge separation process (Fig. 2b), indicating the prevalence of efficient charge generation in the Ternary (H) blend due to the ultrafast 100 fs kinetics. Moreover, compared to D1:A1 blend, the dynamics of charge carriers demonstrates a much faster generation after 100 fs for the Ternary (H) blend (Supplementary Fig. 7). Furthermore, an almost similar charge carrier dynamics was observed after the initial 0.5 ps for the Ternary (H) and the D1:A1 blend. These results further confirm that efficient charge generation in the Ternary (H) blend results from the ultrafast charge separation process. To confirm the consistency of the charge separation in the transients, we investigate the same kinetics in the working OPV diodes under reverse bias. When free charge carriers are extracted by the reverse field, the number of charge carriers decreases and so does the subsequent PIA amplitude. As

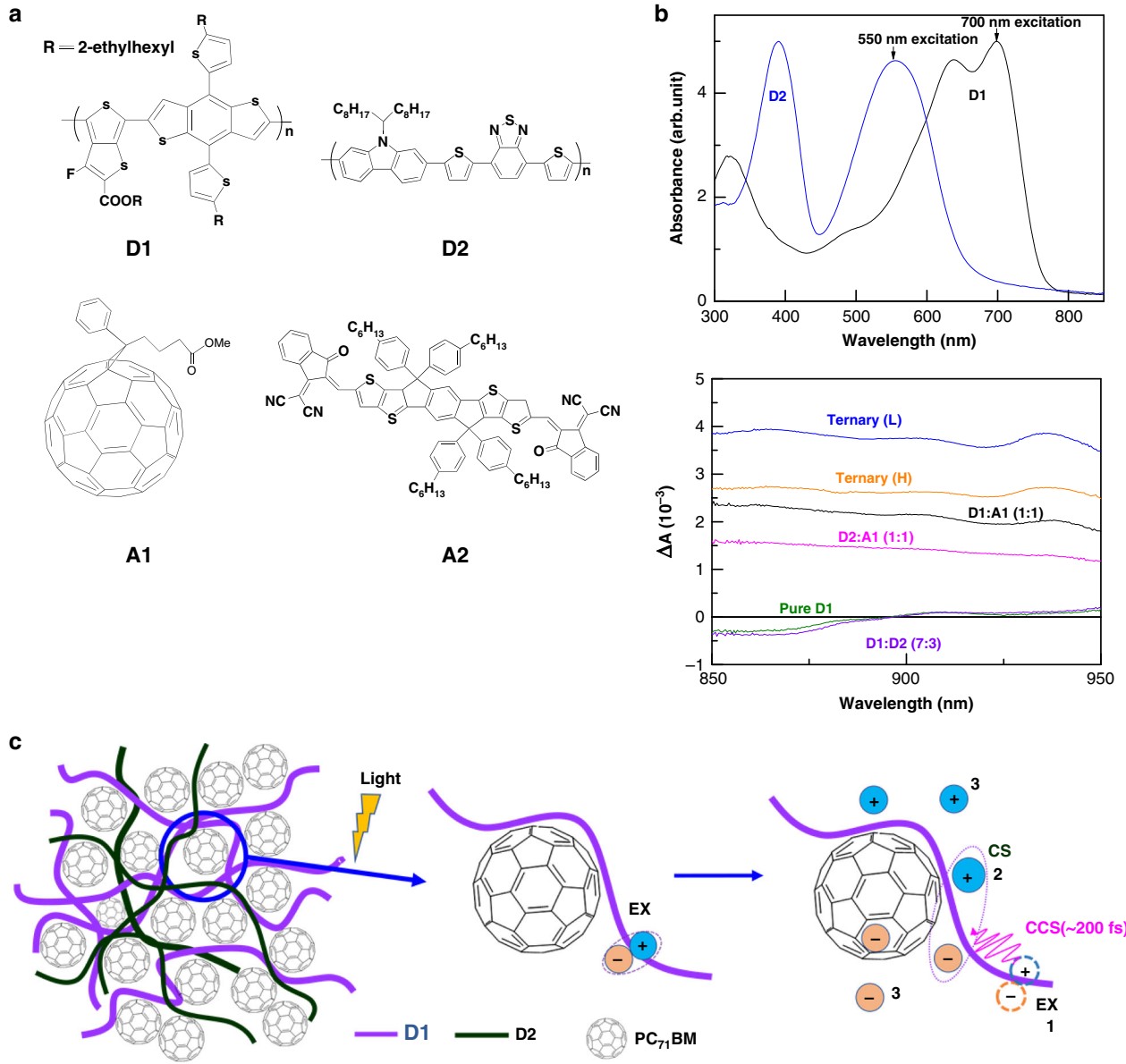

**Fig. 1 Molecular structures and schematics of interfacial photophysical processes. a** Molecular structures used in this study. **b** UV–Vis absorption for PCDTBT (D2, blue solid) and PTB7-th (D1, black solid); Transient absorption signal (~10 ps) from the investigated pristine and blend films, pumped at 550 nm. **c** Simple cartoon illustration of the charge photo-generation at a heterojunction. Light absorption generates excitons (EX) at the interface sites (1), then undergo coherent charge separation (CCS, persist in first 200 fs) into loosely bound electron-hole pairs ((2), charge separated states (CS)). The electron and hole separate further and form free charges (3).

expected, the Ternary (H) device demonstrates a significant bias dependence in the initial ~100 fs timescale (Fig. 2c). In contrast, this feature is not observed in the D1:A1 device (Supplementary Fig. 8).

The rapid exciton dissociation and efficient charge generation in the Ternary (H) blend can also be confirmed by other transient results. Time-resolved photoluminescence (TRPL) is sensitive to the singlet exciton dynamics. In these measurements, we have used the excitation at 445 nm, close to that for the TA measurements (550 nm excitation). The results show that the excited states in the Ternary (H) blend quenches faster (instrument limited) and more completely compared to the other two blends, where PL decay is slower and exhibits a long-lived visible tail up to hundreds of picoseconds (Supplementary Fig. 9). The TRPL measurements exhibit characteristic features of pulse-limited population of the excited state followed by its

depopulation in picoseconds in qualitative agreement with the TA dynamics (Supplementary Figs. 2, 9).

Ultrafast charge separation in the efficient Ternary (H) can be further confirmed by monitoring photoconductivity using time-resolved terahertz (THz) spectroscopy (TRTS). Photoconductivity is a direct measure of the product of charge concentration and mobility brought about by the pump excitation[18]. For all the investigated films here, the same TRTS setup with a time resolution >100 fs and identical experimental conditions have been used. The change in photoconductivity per photon absorbed by the D1:A1 and Ternary (H) blends excited at 400 nm as a function of pump-probe delay is shown in Fig. 2d. Both D1:A1 and Ternary (H) blend exhibit an instrument limited rise time, while for D2:A1 no photoconductivity is obtained. The absence of the photoconductivity signal in D2:A1, despite the TA results showing charge generation before 100 ps (Fig. 2b), implies that

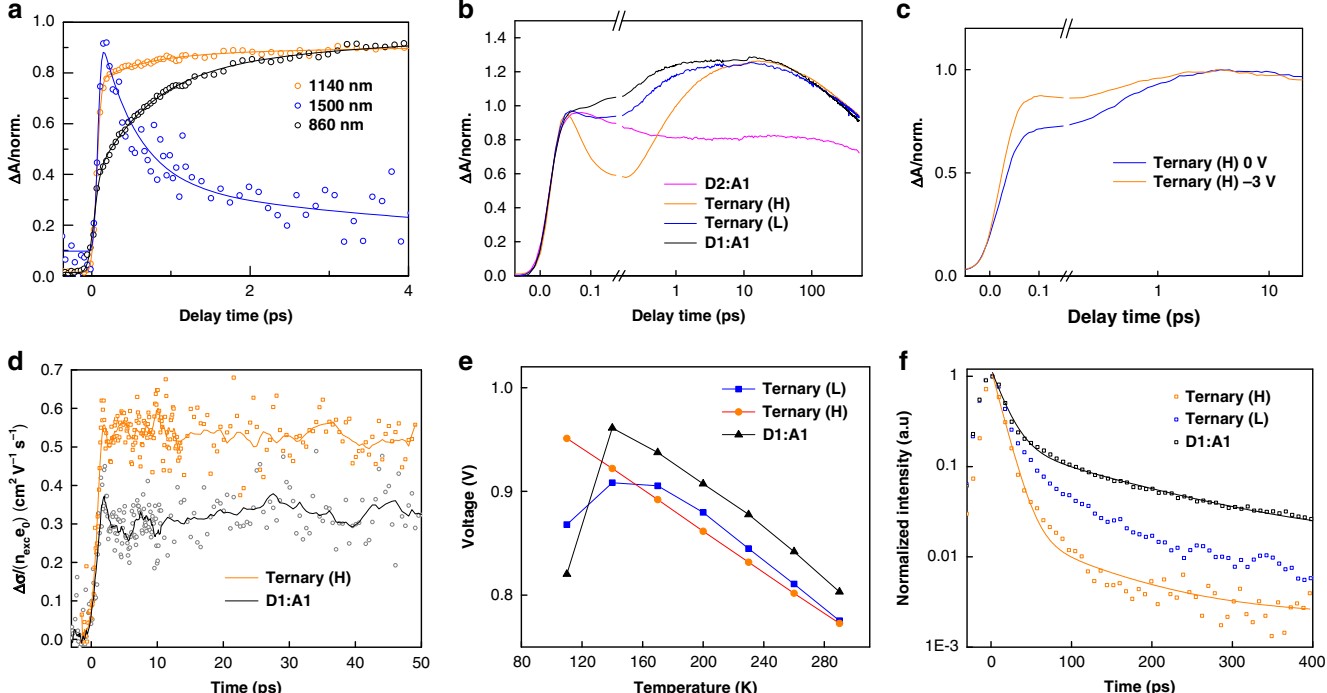

**Fig. 2 Excited-state dynamics of blends. a** Kinetics of Ternary(H), pumped at 700 nm, probed at 860 nm, 1142 nm and 1500 nm, respectively. **b** Kinetics of two ternary blends and two reference binary blends, normalized to the simultaneously acquired signal, pumped at 550 nm and probed at 860 nm. **c** Transient absorption spectra under negative bias of Ternary (H) device, pump at 700 nm and probe at 860 nm. **d** Transient THz photoconductivity kinetics ($\lambda_{exc} = 400$ nm, $I_{exc} = 4.5 \times 10^{12}$ ph/cm$^2$). **e** Temperature dependent photovoltage evolution for three blend diodes. **f** Kinetic curves derived from the time-resolved photoluminescence data by integrating over the spectral range 740–950 nm, pump at 700 nm.

the photo-generated species in the $D2{:}A1$ blend has very low mobility that is below the detection limit. The mobility of ~0.55 cm$^2$/Vs estimated for the Ternary (H) blend is higher than in the $D1{:}A1$ blend (0.3 cm$^2$/Vs), implying that the species present at this timescale (up to 50 ps) are relatively mobile and charged. The absence of photoconductivity signal in $D2{:}A1$, and the lower content of $D1$ in Ternary (H) compared to $D1{:}A1$ blend, are strong indicators that higher photoconductivity in the Ternary (H) blend arises from $D1$. Furthermore, this also means there should be an ultrafast charge generation preceding the time scale that TRTS can access (<100 fs), resulting in the increased photoconductivity in the Ternary (H) blend. These results are consistent with the TA kinetics and confirm that mobile charged species are generated through an ultrafast charge separation in the Ternary (H) blend. Notice that the photoconductivity traces of $D1{:}A1$ and Ternary (H) blend do not show any decay up to 50 ps. This also means that charge carriers can be efficiently extracted and collected as photocurrent, thus directly contributing to the enhanced performance of the Ternary (H) device.

We have also studied the steady-state charge dynamics, following the temperature dependent $V_{oc}$ of the three different devices, as illustrated in Fig. 2e. Since all generated charge carriers recombine at $V_{oc}$ and no long-range charge transport issue is involved, we measure the $V_{oc}$ to exclusively focus on the influence of charge separation on the performance of the device. Here, the resulting charge generation is entropy controlled on the picosecond time scale[19,20], which is consistent with timescale observed in our TRTS measurements. For the Ternary (H) device, the $V_{oc}$ keeps increasing with decreasing temperature (to 110 K), which is significantly different from the other two devices. Note that the calculated charge carrier density shares the same tendency with the $V_{oc}$, as a function of temperature[21,22]. This linear feature indicates an efficient charge separation in the Ternary (H) device, in agreement with our transient results.

A previous study has demonstrated that ultrafast charge separation at the donor/acceptor interface can be influenced by the fullerene aggregate size[23,24]. Our grazing-incidence small-angle X-ray scattering results (Supplementary Fig. 10) demonstrate that the fullerene aggregate size and order are similar in these three different blends. Furthermore, we also rule out the possibility that varied interfacial areas between donor and acceptor[25] contribute to the differences, as comparable interfacial areas are observed in the three devices (Supplementary Fig. 11).

Based on quantum theoretical modeling, it has been proposed that the dynamic disorder due to electronic–vibrational interactions always increases the charge separation barrier; in contrast, static disorder makes the high-energy CS states more accessible[26,27]. We evaluate the static disorder by experiments using a detailed analysis of temperature dependent space-charge limited current (SCLC) (Supplementary Fig. 12). Based on the simulations, we find that the energetic disorder $\sigma$ value for holes is comparable in the three different devices (Supplementary Fig. 13).

These results demonstrate that the differences in morphology, interface area and disorder cannot explain the differences we have observed among the three blends, and therefore a novel mechanism must account for the ultrafast charge generation in the high-performance ternary blend.

**Coherence in photocurrent generation.** Similar to the high excitation energy (550 nm) results, there is an efficient exciton dissociation in the Ternary (H) blend when we selectively excite $D1$ (700 nm) (Fig. 2f, Supplementary Fig. 9). Furthermore, as the rates of free charge generation (Supplementary Fig. 7) and bias dependence (Supplementary Fig. 8) have the same results as under 550 nm excitation, it is rational to assume a participation of the CS state in the free charge generation process under 700 nm excitation. This result indicates that the fast charge separation

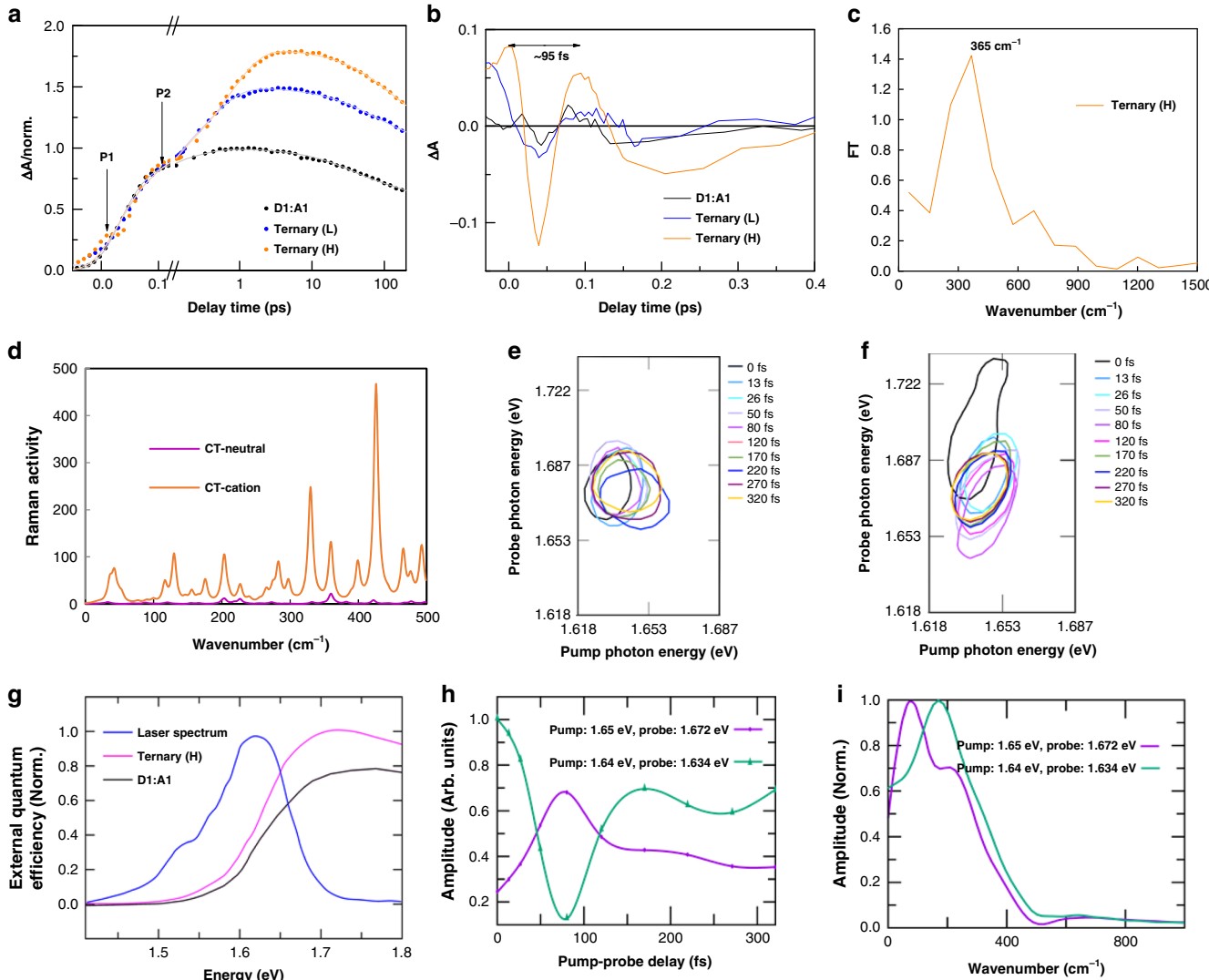

**Fig. 3 Coherent dynamics of blends. a** Kinetic process of the three blends, pumped at 700 nm and probed at 860 nm, normalized at around 100 fs. The charge separation process occurred before 100 fs, followed by the charge generation process. The oscillation signal of Ternary (H) as indicated by the arrow. **b** Corresponding oscillation signal. **c** Fourier transformed spectrum reveals the frequency of the oscillatory components at 365 cm$^{-1}$. **d** Raman spectra calculated at the B3LYP/6-31+G(d) level of theory for the neutral and charged CT conformer. **e** A zoomed version of the 2DPS peak shifts (contour at 90% of the maximum) for the $D1:A1$ and **f** Ternary (H) devices. The color indicates the time delay between the pump and the probe. **g** EQE of the two devices as a function of excitation energy, and the excitation spectrum of the laser. **h** Quantum beats observed at two selected positions in the 2DPS spectra of the Ternary (H) device and **i** the corresponding Fourier transforms.

process is excitation-energy independent, unlike the case which involves molecular ordering and a dependency of the energetic landscape[16].

In particular, the initial rise in the kinetics of the Ternary (H) blend shows a pronounced oscillatory behavior (Fig. 3a), with an amplitude that is significantly higher than in the other two blends (Fig. 3b); these measurements have been done under the same excitation intensity. We have analyzed the oscillatory components by taking the Fourier transform after subtraction of the slowly varying background. The spectra shows a pronounced broad feature below 500 cm$^{-1}$ encompassing a number of low frequency vibrational modes. The main peak at ~365 cm$^{-1}$ (Fig. 3c) is similar to the ground-state vibrational mode at ~370 cm$^{-1}$, detected by resonance Raman spectroscopy of the blends and materials (Supplementary Fig. 14). For the Ternary (H) blend (Supplementary Fig. 7), a significantly increased charge carrier generation rate was observed after the initial 100 fs; this result is consistent with the charge transfer excitons kinetics (Fig. 3a), also

demonstrating a significantly enhanced generation rate after the initial 100 fs, and indicates that the initial 100 fs oscillatory feature is coupled with the ultrafast charge separation process. Furthermore, the oscillation persists ~200 fs, longer than the ultrafast charge separation process. This result is different from the high frequency mode reported elsewhere[9]. The correlation between charge transfer excitons (and/or charge carrier) density and oscillation amplitude indicates that the excited-state vibrational mode with a frequency of ~365 cm$^{-1}$ drives the charge separation process. We have analyzed the vibrational mode in detail considering a model system of $D1$ based on quantum theory (Fig. 3d, Supplementary Fig. 15). The main peak around 360 cm$^{-1}$ is in agreement with the experimental results obtained from the Fourier transform of the TA and from the Raman spectra. The vibrational mode can be attributed mainly to an in-plane breathing of the benzo[1,2-b;4,5-b']dithiophene donor unit in $D1$ (Fig. 4a; Supplementary Movie 1).

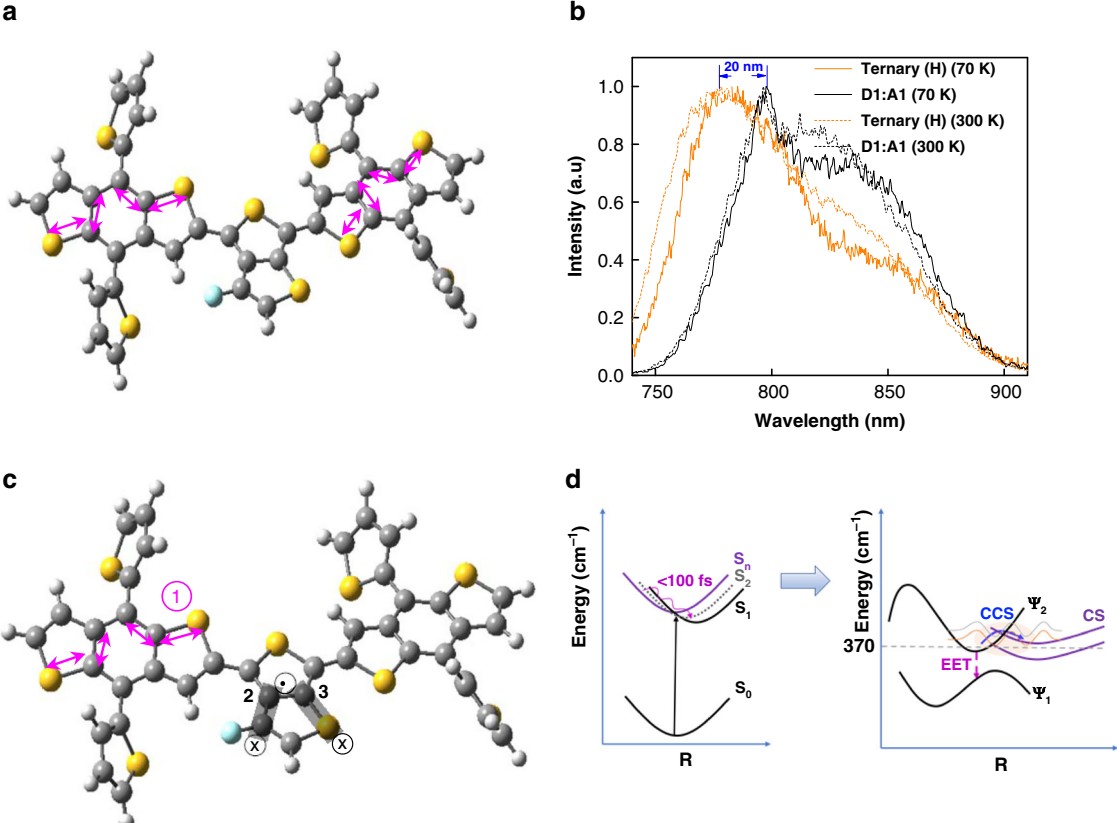

**Fig. 4 Coherent kinetic process. a** The in-plane breathing mode. The initial excitation delocalized by the in-plane breathing mode. Vibration direction indicated by arrow. **b** The transient photoluminescence peak of the Ternary (H) demonstrates a blue shift compared to the D1:A1 blend at the first 20 ps timescale. **c** Out-of-plane mode (gray shadow). For the 200 cm$^{-1}$ vibration mode, mainly due to the out-of-plane vibration of 3-fluorothieno[3,4-b] thiophene by the $C_2$ and $C_3$ bond. The in-plane breathing mode of the benzo[1,2-b;4,5-b']dithiophene① simultaneously with the out-of-plane vibration of 3-fluorothieno[3,4-b]thiophene, aid in charge separation simultaneously during the first 200 fs. Vibration direction indicated by arrow. **d** Kinetic approach for excitation energy transfer (EET) and coherent charge separation (CCS) process. Here, the resulting charge separated states (CS) are delocalized. Potential energy surfaces sections (PES) defined by the electronic wavefunctions ($\Psi_1$, $\Psi_2$) of the excited states as a function of torsional, ring-breathing and intra/intermolecular bond stretching coordinates(R). The gray and pink line represent the EX and CS vibronic wavefunction, extended on both PES.

We have further analyzed the Raman activity corresponding to the in-plane breathing mode. Compared to the neutral, the positively charged CT conformer demonstrates a significant enhancement of the Raman activity (Fig. 3d, Supplementary Fig. 15). This result indicates the breathing mode is in resonance with the ultrafast charge transfer process, which results in efficient CS states formation, consistent with our experimental results. Thus, the initial excitation can be dissociated into CS states through this specific breathing mode.

Our transient results demonstrate that the ultrafast process from EX to CS is temperature independent (Supplementary Fig. 5), and the corresponding oscillation signal dephases quickly (in <200 fs, consistent with the recently resolved electronic coherence lifetime[28]). Furthermore, the energy gap between the initial EX (emission peak ~800 nm) and the CS states (emission peak ~780 nm) (Fig. 4b, Supplementary Fig. 16), is resonant with the breathing mode (~365 cm$^{-1}$). Thus, the resulting electron–vibrational (vibronic) coupling, via electronic states coupled to the in-plane breathing mode, plays the most effective role in promoting CS formation when it is resonant with the transition energy[29]. These results further indicate that the electronic coherence may also contribute to the ultrafast charge separation in the Ternary(H) blend. Note here that the product CS states are located at higher energy than the primary EX states. This uphill energy process thus utilizes coherence, breaking the energy barrier between these two states, which is unlike the

normal downhill hopping case[30]. Considering that the coherence process lives as long as the breathing period lasts, electronic coherence could be enhanced by this vibrational coherence.

In order to further conclusively prove that the coherent motion is involved in the charge transfer process and the subsequent photocurrent generation, we have used 2D coherent photocurrent spectroscopy (2DPS)[31–33]. This method is an extension of 2D electronic spectroscopy, but uses photocurrent generated in a device as the measured signal when illuminated by two pairs of pulses of light in a pump and probe setup for following the nonlinear response of the excited photodiode (see Methods section). In distinction to our method for nonlinear imaging of photocurrent for identifying recombination mechanisms[34,35], where the amplitude of the nonlinear incoherent response is used, here the phase of the photocurrent is extracted by varying the interpulse time delays. This has been shown to account for the coherent nonlinear interactions in this type of spectroscopic measurement[36,37]. For all the investigated devices here, the same setup with a temporal resolution of about 15 fs and identical experimental conditions have been used. The spectra at the zero time delay between the pump and probe shows that the main peak is shifted above the diagonal (Fig. 3e, f, Supplementary Fig. 17). There are different mechanisms, such as many body effects[38] and AC (optical) Stark effects[39] that cause the shifts. Among the two effects, the upshift in the probe frequency indicates that the AC Stark effect plays the dominant role. The AC Stark effect can lower as well as increase the band gap

of the material depending on the excitation spectrum[40]. In our experiment, most of the excitation spectrum (Fig. 3g) is below the band gap for both the devices. In this case, the AC Stark shift increases the band gap, which is observed as a shift in the 2D peak above the diagonal. In order to further analyze the 2D spectra, we show the evolution of the peak position as contour lines at 90% of the maximum amplitude for the $D1$:$A1$ and Ternary (H) devices in Fig. 3e, f, respectively. After the pump-probe overlap (i.e., beyond 15 fs), the main peak shifts to the diagonal. The peak position in both the samples is at 1.653 eV (750 nm), which corresponds to the excitonic transition. Note that the rise in the amplitude with energy is gradual compared to most of the semiconductor systems. This indicates that other bound states, such as polarons, also contribute to the photocurrent. Nevertheless, their contribution is insignificant compared to the excitons and excitations above the bandgap. Beyond the pulse overlap, we observe a significant oscillation of the peak position of the Ternary (H) device within 200 fs. Then the peak position remains static. Such transient feature is not observed in the $D1$:$A1$ device (Fig. 3e). Compared to a pristine molecular system, the polymer blend is heterogeneous. A number of vibronic states from the different components in the blend contribute to the spectral features above the bandgap. As a consequence, the excitation above the bandgap results in a rather constant EQE without sharp spectral features (Fig. 3g), which makes it difficult to infer coherences as distinct off-diagonal peaks in the 2D spectra. This is similar to the continuum of states in a semiconductor. Nevertheless, the coherences can be inferred from the oscillations in the off-diagonal positions of the broad spectral lineshape. In order to quantitatively analyze the oscillation frequencies, we monitor the quantum beat signals at two positions on the off-diagonal as shown in Fig. 3h. In the first position, pump is at 1.65 eV and probe is at 1.672 eV and in the second position, pump is at 1.644 eV and probe is at 1.634 eV. The oscillations in these two positions are in anti-phase. The Fourier transforms show significant vibrational contribution to the photocurrent at frequencies below 500 cm$^{-1}$ (Fig. 3i). The main peak in the spectra is around 200 cm$^{-1}$, which is different from the main peak in the TA results. This result can be rationalized based on the fact that in the detected 2D photocurrent, the results are weighted by the EQE of the photocurrent, while in the TA we only measure the optical response. Note that the window of oscillation frequencies (Fig. 3i) includes the vibrational mode that is resolved by TA. Similar to the TA results, the oscillations dephase quickly (~200 fs). Our results show that the low frequency vibrational modes have greater contribution to the coherent photocurrent generation. The low vibrational mode (~200 cm$^{-1}$) can be mainly attributed to the out-of-plane vibration of the 3-fluorothieno[3,4-b]thiophene donor unit in $D1$ (Fig. 4c, Supplementary Movie 2 and Movie 3). Such out-of-plane conformer hinders close main-chain stacking and influences the local configuration of the $A1$ moieties near the $D1$ backbone[41,42], facilitates the intermolecular charge transport between $D1$ and $A1$[43]. Note here that both vibrational modes (~365 and 200 cm$^{-1}$) aid in charge separation simultaneously and persist in the first 200 fs. The results clearly indicate a coherent charge transfer process within 200 fs of the excitation, that has a significant contribution to the photocurrent in the Ternary (H) device.

It has been suggested that there is strong coupling between electronic potential energy and the torsional mode, which facilitates the initial coherent excitation energy transfer (EET) (<100 fs) and charge separation process[13,14]. Transient results demonstrate that the simultaneous process of EET from the excited state is significant in the Ternary (L) and $D1$:$A1$ blend, compared to the Ternary (H) blend (Supplementary Fig. 19). Due to efficient charge separation, the competing initial EET process is suppressed in the Ternary (H) blend. Our study indicates that an initial planarization from a torsional mode, is not the first decisive step in this ternary blend.

Here, the coherent charge separation (CCS) outcompetes the electronic relaxation and results in CS products (Fig. 4d).

To generalize the contribution of the ultrafast charge separation in OPVs performance, we also study blends with a non-fullerene acceptor (Supplementary Fig. 20). ITIC has planar and rigid conformation and has been widely used in high efficiency donor/acceptor blends[44]. The rigidity suppresses torsional relaxation, and thus improves efficient EET process. TA demonstrates that, in the initial time scale (100~500 fs), EET is significant in both binary and ternary blends. Thus, the competing charge separation is suppressed, and resulting in similar device performance (Supplementary Fig. 21).

## Discussion

The primary charge separation process in this organic photovoltaic blend occurs by vibronic coherence and is controlled by the electronic and vibrational coupling unique to the low frequency in-plane breathing mode and out-of-plane vibration mode. Our study shows that even if the coherence time is short (~200 fs), this process is helpful in reducing charge–pair Coulomb interaction and efficiently generating charged species, thereby contributing to the photocurrent enhancement. Specifically, the breathing modes found here can work as an effective design building block, giving clear inspiration for designing chemical structures that exploit coherent quantum mechanical effects in organic materials and nanostructures. Coherence phenomena have been proposed to contribute in some biological photosynthetic systems[29,45–47], and some reviews have also proposed that coherence might enhance charge transfer[46,48]. Very recently, coherence was found to play a key role for the near-unity CS efficiency in a natural reaction center complex[28]. To explore the next-generation of photovoltaic materials, new design strategies based on understanding the working mechanism are needed. We anticipate that our identification of a dynamic contribution from vibronic coherence can be used to design new molecules and materials for improved charge generation.

## Methods

**Sample preparation**. PTB7-th and ITIC were purchased from Solarmer Material Inc, Beijing, China. PCDTBT were purchased from 1-Material Inc, Canada. PCDTBT with low and high molecular weight is Mw 64 kDa and 450 kDa, respectively.

Two common polymers PTB7-th (**D1**), (Poly[4,8-bis(5-(2-ethylhexyl)thiophen-2-yl)benzo[1,2-b;4,5-b′]dithiophene-2,6-diyl-alt-(4-(2-ethylhexyl)-3-fluorothieno [3,4-b]thiophene-)-2-carboxylate-2-6-diyl)]) and PCDTBT (**D2**), (poly[N-11W−henicosanyl-2,7-carbazole-alt-5,5-(40, 70-di-2-thienyl-20, 10, 30-benzothiadiazole)]), were used as electron donors. The PC$_{71}$BM (**A1**), (phenyl-C71-butyric acid methyl ester) and small molecule ITIC[44] (**A2**), (3,9-bis(2-methylene-(3-(1,1-dicyanomethylene)-indanone))-5,5,11,11-tetrakis(4-hexylphenyl)-dithieno[2,3-d:2′,3′-d′]-s-indaceno[1,2-b:5,6-b′]dithiophene), were used as acceptors in this study. PCDTBT is a widely studied noncrystalline polymer, with a planar structure in the excited state[49]. When PCDTBT is blended with PCBM, internal quantum efficiency close to 100% was obtained[49,50]. By using different molecular weights and optimizing the donor phase loading ratios in the ternary **D1**:**D2**:**A1** blend with respect to the binary blend, a substantial improvement in short-circuit current density ($J_{sc}$) and fill factor (FF), which resulted in a substantially increased power conversion efficiency of 9.5%. We focus on two optimized ternary blends (**Ternary (L)**, $D1$:$D2$(L):$A1$ = 9:1:10, with low molecular weight in $D2$; **Ternary (H)**, $D1$:$D2$(H):$A1$ = 7:3:10, with high molecular weight in $D2$) and two reference binary blends (**D1:A1, D2:A1**) (Supplementary Fig. 1). Absorption and photoluminescence spectra, external quantum efficiency (EQE) and current ($J$)–voltage ($V$) curves are provided in Fig. 1 of the Supplementary Information.

Binary and ternary devices were prepared in the normal device configuration ITO/PEDOT:PSS (40 nm)/active layer (100 nm)/LiF(0.6 nm)/Al(90 nm). PEDOT: PSS (Baytron P VP Al 4083) was spun-cast and baked at 140 °C. The active layer was spin-coated from a 10 mg mL$^{-1}$ chlorobenzene solution. LiF (0.6 nm) and Al (100 nm) were used as cathode, was thermally evaporated under vacuum (<10$^{-7}$ torr). Hole only devices were prepared in the device configuration ITO/PEDOT: PSS/active layer/MoO$_3$/Al.

For indication case, in the main and Supplementary Materials text, the optimized blends are **Ternary (H)** (PTB7-th:PCDTBT(H):PC$_{71}$BM (7:3:10),

PCDTBT with high molecular weight) and **Ternary (L)** (PTB7-th:PCDTBT(L): PC$_{71}$BM (9:1:10), PCDTBT with low molecular weight).

**Low temperature measurement**. Normal devices and single-carrier devices were mounted in a liquid-nitrogen cryostat for temperature-dependent measurements. The J–V curves were measured using a Keithley 2400 source meter at temperatures ranging from 90 to 300 K. The temperature was monitored and controlled using a LakeShore 330 Autotuning Temperature Controller. The light source was a blue laser (532 nm), and the light intensity was tuned to give a $V_{OC}$ similar to one-sun conditions at room temperature for Voc-T test. Photoluminescence (PL) and electroluminescence (EL) spectra were detected using a light guide positioned close to the cryostat window. The emission detection system was a Newton EM-CCD Si array detector cooled at −60 °C in conjunction with a Shamrock sr 303i spectrograph from Andor Tech.

**Time resolved transient absorption (TA) instrumentation**. Transient absorption (near-infrared) measurements were performed by means of regeneratively amplified, mode-locked Yb:KGW (Ytterbium-doped potassium gadolinium tungstate) based femtosecond laser system (Pharos, Light conversion) operating at 1030 nm and delivering pulses of 200 fs at a 1 kHz repetition rate. This laser is then used to pump two non-collinearly phase-matched optical parametric amplifiers (NOPAs, Orpheus-N, Light Conversion). A first one was used to generate pump pulses centered at 540 nm with a pulse duration of 35 fs. The second NOPA was used to generate probe pulses of about 40 fs duration in the spectral region from 850 to 960 nm for differential absorption measurements. The probe was time delayed with respect to the pump by a mechanical delay stage.

Transient absorption measurements (infrared) were performed using a Helios setup. The transient dynamics in fs–ns time region (50 fs–7 ns) was acquired by Helios that works in a nondegenerate pump–probe configuration. The pump pulses were generated from an optical parametric amplifier (OPerA Solo) that was pumped by a 1-kHz regenerative amplifier (Coherent Libra, 800 nm, 50 fs, 4 mJ). A mode lock Ti-sapphire oscillator (Coherent Vitesse, 80 MHz) was used to seed the amplifier. For 800 nm pump, the laser from the regenerative amplifier was directly used. The probe pulses were a white light continuum generated by passing the 800 nm fs pulses through a 1 cm sapphire plate for the infrared part (840–1600 nm).

The measured sample solution was spin coated on the quartz substrates and mounted in a nitrogen cryostat. A consistency results of the near-infrared signal was achieved from the two different transient absorption setups.

**Time resolved terahertz (THz) instrumentation**. The photoconductivity kinetics were measured by a setup similar to that in the ref.[2]. Laser pulses (796 nm, 80 fs pulse length, 1 kHz repetition rate) were generated by a regenerative amplifier (Spitfire Pro XP, Spectra Physics) seeded by a femtosecond oscillator (MaiTai, Spectra Physics). The laser beam was split into three. The first beam (400 μJ/pulse) was used to generate THz radiation by optical rectification in a MgO:LiNbO$_3$ crystal. The second beam was used for electro-optical sampling of the THz pulses in a (110) ZnTe crystal. The third beam, which was used for excitation was converted to the second harmonic (398 nm) in a BBO crystal. To avoid absorption of the THz radiation by water vapor, the setup was purged with dry nitrogen. For all the investigated films in the text, the same time-resolved THz spectroscopy (TRTS) setup and identical experimental conditions were used.

The primary result of TRTS is the change in conductivity (Δσ) of the material[51] following optical excitation as expressed by the equation:

$$\Delta\sigma = \xi \times (\mu_e + \mu_h) = -\frac{\Delta E_{exc}(\omega)}{\Delta E_{gx}(\omega)} \frac{c\varepsilon_0}{Fe_0} \frac{1}{1 - e^{-\alpha L}}, \qquad (1)$$

where ξ is quantum yield of charge generation, $\mu_e$ and $\mu_h$ are the electron and hole mobility, respectively, $\Delta E_{exc}$ is the THz electric field transmitted through the sample after photo excitation while $\Delta E_{gs}$ is the ground state THz electric field, $\varepsilon_0$ is permittivity of vacuum, c is velocity of light, F is the fluence in ph cm$^{-2}$, $e_0$ is the elementary charge, α is the absorption coefficient, and L the thickness of the sample. The quantity that is obtained from the above equation has the unit of mobility in cm$^2$ V$^{-1}$ s$^{-1}$.

**2D photocurrent spectroscopy**. The schematic of the experimental setup is shown in the Supplementary Fig. 17. In the setup, a chirp pre-compensated pulsed laser beam (about 70 MHz repetition rate, 15 fs pulse duration and center wavelength of 790 nm) from a Ti-Sapphire oscillator (Synergy from Femtolasers) is split into two. Each beam goes through a Mach-Zehnder interferometer (MZ). Acousto-optic modulators, AOM 1, 2, 3, and 4 on the arms of the interferometers MZ1 and MZ2 modulate the phases of the beams at frequencies $\phi_1$, $\phi_2$, $\phi_3$, and $\phi_4$, respectively. The delay line DL1 controls the time delay τ between the first two pulses (pulse 1 and pulse 2). Similarly, DL3 controls the time delay t between the last two pulses (pulse 3 and pluse 4). The delay T between pulse 2 and pulse 3 is controlled by DL2. All the beams after MZ1 and MZ2 are recombined using a 50/50 beam splitter. One of the outputs from the beam splitter is used to excite the sample S, while the other output goes to a monochromator MC. The monochromator selects a narrow spectrum of the light, which provides the reference detected by a Si-photodiode. The photocurrent signal from the sample is amplified by a

preamplifier (SR570, Stanford Research Systems). The signal and the reference are digitized simultaneously at the rate of 10 MSa/s (mega samples per second). The phases and amplitudes of the signals modulated at the frequencies $\phi_4 - \phi_3 - \phi_2 + \phi_1$ and $\phi_4 - \phi_3 + \phi_2 - \phi_1$ are extracted from the digitized data by using the algorithms of the generalized lock-in amplifier (GLIA)[52–54]. The measurements are done in time domain by varying the time delays. The spectra in the frequency domain are obtained by the Fourier transforms along τ and t for each time delay of T. The Fourier transform along τ gives the pump frequency and the Fourier transform along t gives the probe frequency for the different waiting times (population times) T. The real part of the 2D spectra obtained from the signals at the two frequencies are summed to obtain the total correlation spectra, which are shown in Supplementary Fig. 17.

**Vibrational modes calculation based on quantum theory**. Methodology: Optimization and Raman spectra has been calculated at the B3LYP/6-31+ G(d) level of theory using the Gaussian Package.

**Cis trans conformation between the small and large units**. Model systems: To investigate the conformational space between the large and the small unit of PTB7-th, we build the four following models comprising a small unit between two large units in combination of cis and trans position.

## Data availability
The datasets generated during and/or analysed during the current study are available from the corresponding author on reasonable request.

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

## Acknowledgements

We thank Dr. Wei Ma for performing grazing-incidence small-angle X-ray scattering measurements. This work has been supported by the Knut and Alice Wallenberg Foundation (KAW) through a Wallenberg Scholar grant to O.I. A.Y. acknowledges support of the KAW and Crafoord Foundations. W.M.C. thanks the Knut and Alice Wallenberg Foundation for support (KAW 2014.0041). Q.B. acknowledges the China Scholarship Council (CSC)(no.201508320244).

## Author contributions

Q.B. and O.I. designed the project. Q.B. performed the device-based tests and partici-pated in all experiments. F.M. and Q.W. carried out the transient absorption measure-ments. S.C. carried out the transient photoluminescence measurements. X.S. carried out the transient terahertz measurements. K.J.K. carried out the 2D photocurrent measure-ments and analysis. M.L. carried out the DFT calculations. I.B., W.C., C.P. and A.Y. participated in data interpretation. Q.B. analyzed the data and wrote the manuscript together with O.I. All authors discussed the results and commented on the final manuscript.

## Competing interests

The authors declare no competing interests.
