## [Peer Review File · Nature Communications]

Reviewers' comments:

Reviewer #1 (Remarks to the Author):

The authors study charge transfer dynamics in binary and ternary organic semiconductor blends comprising two polymer donors and a fullerene acceptor. They argue that "coherence plays a role in the initial ~ 200 fs" charge transfer dynamics.

To support this claim, they report time-resolved transient absorption spectra and 2D photocurrent spectra. In the transient absorption spectra, they excite the donor absorption band and probe at energies below the donor absorption band, where typically photoinduced absorption of polaron-pair or polaron excitations can be found. Transient absorption spectra in this region are measured (Fig. 1b) and related to "charge separated states". A clear assignment of the PIA band, however, is lacking, so that is not easy to judge what is actually probed in these measurements.

The PIA dynamics in the ternary blends show some decay on a ~ 50 fs time scale, and this is taken as a signature of an ultrafast charge separation process. Since it is not really clear which states are actually probed, it is difficult to say whether these dynamics indeed related to charge separation process. If the probe laser indeed probes charge-separated states (polaron-pair or charge-transfer excitons), then I would rather expect a continuous rise of the PIA amplitude, as recently demonstrated, for example, in pump-probe and 2DES studies of polythiophenes and polythiophene-fullerene blends. The argument that "intermediate CS state helps in free charge generation" (p.5) is therefore not sufficiently supported by the pump-probe data. The authors conclude that this ultrafast decay is consistent with ballistic charge motion. Even though this might be possible, I do not understand how meaningful conclusions about ballistic charge motion can be drawn from a decay of a pump-probe signal alone, in particular if it is not clear which states are probed.

The authors go on by stating that the pump-probe dynamics in Fig. 3a show "a pronounced oscillatory behavior". Honestly, I do not see this oscillatory behavior. Even if I accept the subtraction method used by the authors to extract the "oscillation signal" in Fig. 3b, this signal does not show "clear oscillations". (I must admit that I am very skeptical about this subtraction. I do not see much difference between the early time dynamics of the three samples. Nevertheless the data in Fig. 3b for the three samples look very different.). I see a negative peak at 45 fs and a positive peak at 100 fs in the red curve in Fig. 3b. These peaks may indeed arise from an oscillatory motion with a period of 95 fs. At later times, however, this oscillatory mode is no longer seen in the data. Obviously, more and less noisy experimental data would be needed to support the claims made by the authors.

Finally, the authors also present 2D photocurrent spectra. The results for a dimer are presented in Fig. 3e and S14g. I understand that the data in Fig. S14 show a single diagonal peak in the 2D spectra at an energy of about 1.64 eV (750 nm). The origin of this peak is not assigned. It seems to lie at the low-energy edge of the donor absorption. From the absorption spectra of the dimer it is unclear why excitation of this energy region should give rise to a peak in 2D spectra. The origin of the peak is not explained by the authors. I therefore understand that it is not understood. The 2D signal in Fig. S14 shows no significant changes when varying the time delay. Fig. 3e shows some (unexplained) contour line representation of this peak. Now, the peak center is shifted to an off-diagonal position – which is obviously inconsistent with the representation in Fig. S14. This discrepancy is not even mentioned in the text.

Now, in Fig. 3f and S14a, similar data are shown for the ternary blend. I must mention that it was extremely difficult to decipher these images since they were all superimposed by a bright green background. I understand, however, from S14a that the data show a similar peak as in S14b. The origin of this peak is again neither discussed nor explained. The amplitude of this peak is plotted in Fig. S14c and shows a similar dip/peak structure as in the pump-probe data in Fig. 3b. Now, however dip (75 fs) and peak (150 fs) occur at substantially different time delays. The claim of the authors that "the time scale of the oscillation is similar to that observed in the transient absorption measurements" is simply not justified. It is also fundamentally unclear to me how these peak variations can be taken as an indication of "a coherent charge transfer process". Coherences in two-dimensional spectra are observed as distinct off-diagonal peaks in the 2D map. Such peaks

are neither observed nor discussed in the present data.

I therefore conclude the data reported in the manuscript do not support the claim expressed in the title of this paper. Publication of those data is clearly premature and cannot be recommended. Additional experimental evidence is needed before claims about coherence in charge separation processes can be made. Such claims must be based on a thorough data analysis – which is not presented in the current work. I therefore do not recommend to publish this work in Nature Communications.

Finally, I want to remark that the discussion of “coherence” that is given in this paper is rather superficial since no distinction between electronic, vibronic or vibrational coherences is made. A more in-depth explanation of what the authors mean by “coherence” should be given. 2D-photocurrent spectra of organic materials have been studied by Bittner et al (and others). References to this work should be added.

Reviewer #3 (Remarks to the Author):

Optoelectronic properties and ultrafast energy and charge carrier dynamics in several bulk-heterojunction organic semiconductor devices have been examined with a number of complementary state-of-the-arts electronic and vibrational spectroscopies to probe appearance of electron-vibrational (vibronic) dynamics. The authors conclude that ultrafast transformation of tightly-bound photoexcited exciton into the charge-separated state on ~ 200 fs timescale is facilitated by a coherent vibronic dynamics associated with the breathing mode on the polymer at ~ 365 cm^{-1} . Such observations add valuable detailed experimental observation into decades-long arguments on how coherent evolution of electronic and vibrational degrees of freedom affect dynamics of carriers and may have consequences for future organic light-harvesting and lighting applications.

Overall, the experimental results are convincing, well presented and the manuscript is well written. Consequently, the article potentially may be published in Nature Communications.

However, the main conclusions of the paper are not strongly supported by the results. Thus, I do not recommend this paper for publication in Nature Communications in its present form. I hope that the authors will be able to address these issues in the revision.

Main critique: The main conclusion of this work is that “The vibrational mode found here can be used to inspire the design of new photovoltaic materials”. This vibration is identified as a breathing mode of D1 donor compound. To me, the central question remains unanswered: why does this vibration promote charge separation in the tertiary H device (D1:D2:A1) while it is not affecting the binary device (D1:A1) performance? What is so special in mixing in D2 compound which does not have such vibrations and has higher lying excitonic states (that seemingly should not participate in charge separation)? The authors convincingly “demonstrate that the differences in morphology, interface area and disorder cannot explain our observed differences among the three blends.” Why does then 7:3:10 mixture activate this breathing mode compared to 9:1:10 and 10:0:10 mixtures for D1:D2:A1 blend?

Other suggestions:

1. Abstract: “The vibrational mode found here can be used to inspire the design of new photovoltaic materials. ” I believe that the authors would like to say that the presented efficient ultrafast charge separation process assisted by specific coherent vibrational motion, can be used to inspire....

2. P. 5: "After 0.5 picoseconds (ps), the kinetic curve represents the deactivation of the lowest-level excited state." Pls clarify if you mean the lowest exciton D1 state.

3. P. 7: "resulting charge generation process is entropy controlled". Clarify, why?

4. P11: "the energy gap between the initial EX (emission peak ~ 800 nm) and the CS 229 states (emission peak ~ 780 nm), is resonant with the breathing mode (~ 360 cm^{-1})." Unclear. This is significant. Why should the resonance between fully relaxed excitonic state and the CS state be important? I do not see any conclusive evidence to make this statement (except the numbers are lining up ok). It seems to me that the transition to CS occurs dynamically before equilibration. And this also should be present in D1:A1 mixture but does not play a significant role.

5. Fig. 1. Panel b) Transient absorption curves are too tiny... these may deserve the entire panel. Panel c) colors in the cartoon for D1 and D2 are too similar to be easily distinguishable.

6. Fig. 3f: appears as a green square on the screen. Pls check the original version.

7. General comments for all Figures: the fonts are small and the lines are too thin to be easily readable. This should be fixed for production version.

8. On the screen presented calculated vibrational modes are not well seen. The pictures look like the molecular views from different angles.

Response to Reviewers' Comments on NCOMMS-19-18303-T

We thank the reviewers for their constructive comments and careful reviews of our manuscript, which have helped us to further improve our manuscript. In the past months, we have taken additional data, redone some analysis, and modified the presentation in our manuscript to address all the reviewers' comments/questions. We believe that the revised manuscript is much improved from the original version and it can now be recommended for publication in *Nature Communications*. In the following, we present our point-to-point responses to all the reviewers' comments and indicate corresponding changes made in the revised manuscript. The major changes have been highlighted by colour in the "revised manuscript with marks".

Response to Reviewer #1's comments:

Reviewer's comment #1:

In the transient absorption spectra, they excite the donor absorption band and probe at energies below the donor absorption band, where typically photoinduced absorption of polaron-pair or polaron excitations can be found. Transient absorption spectra in this region are measured (Fig. 1b) and related to "charge separated states". A clear assignment of the PIA band, however, is lacking, so that is not easy to judge what is actually probed in these measurements.

Response: We appreciate the reviewer's careful reading of the manuscript and the thoughtful comments.

The reviewer's major concern is about the assignment of PIA feature of "charge transfer states" (charge separated states). To address this, we have performed additional extended infrared transient absorption measurements (Results are shown on Figure R1, Figure R2 below and Fig.S4 in Supplementary Information (SI) and Figure. 2a in the revised manuscript).

The infrared kinetics (Figure R2, Fig.2a) demonstrate that the charged species in the 860 nm PIA band increases simultaneously with the singlet exciton dissociation in the initial picoseconds, indicating that the ultrafast charge transfer occurs between the singlet exciton states and the 860 nm PIA band species. The new results strongly support the assignment of PIA band around 860 nm to the

charge transfer states. Thus, we focus on the 860 nm signal, as its kinetics provides clear measurements of charge carrier generation from the initially photogenerated excitons on the polymer. With these new evidences, the reviewer's concerns are well addressed.

In the revised manuscript, we have made revisions accordingly. For the signal assignment and related kinetics difference, the infrared transient results (Figure R1) has been added in Fig.S4 in the Supplementary Information (SI) and additional Figure 2a (Figure R2) has been added in the revised manuscript.

Figure R1. Excited-state dynamics in the infrared region. **a.** Transient dynamics of D1:A1 blend. **b.** Transient dynamics of Ternary(H) blend. For these infrared transient tests, pump at 700 nm. Three main spectral features are detected: i) the band in the spectral range 850-900 nm, that in the pure donor D1 has negative amplitude, is here exchanged to a positive signal, that is rapidly quenched by the formation of the interfacial charge transfer states (CT) PIA. ii) The PIA peaked at 1140 nm assigned to charge carrier absorption, that grows in the first 300 fs and iii) the PIA band peaking at 1500 nm assigned to excited singlet absorption, that shows a decay that matches both CT and charge carrier PIA rise.

Figure R2. Kinetics of Ternary(H), pumped at 700 nm, probed at 860 nm, 1140 nm and 1500nm, respectively. A fast excited singlet decay (1500nm) that matches both charge transfer states (860 nm) and charge carrier (1140 nm) PIA rise.

Reviewer's comment #2:

The PIA dynamics in the ternary blends show some decay on a ~ 50 fs time scale, and this is taken as a signature of an ultrafast charge separation process. Since it is not really clear which states are actually probed, it is difficult to say whether these dynamics indeed related to charge separation process. If the probe laser indeed probes charge-separated states (polaron-pair or charge-transfer excitons), then I would rather expect a continuous rise of the PIA amplitude, as recently demonstrated, for example, in pump-probe and 2DES studies of polythiophenes and polythiophene-fullerene blends. The argument that "intermediate CS state helps in free charge generation" (p.5) is therefore not sufficiently supported by the pump-probe data.

Response: With the strong support of our new experimental results (Figure R1 and Figure R2), we can safely assign the probed 860 nm species to the charge-transfer excitons. By exciting D1 only (700 nm), a continuous rise of PIA amplitude is observed (Figure R3, and Fig.S7 in Supplementary Information (SI)), as reviewer pointed, which is consistent with other pump-probe and 2DES studies of polythiophene-fullerene blends. The ~ 50 fs decay process that is observed by using 550 nm pump is assigned to the ultrafast charge transfer from excited polymer to PCBM. As a result, a bound electron-hole pairs is formed with the electron located on PCBM and the hole located on polymer, as displayed by the subsequent rise signal (Figure.2b, 0.3-10 ps). Compared to reference D1:A1 blend (a continuous rise of PIA), the subsequent rise demonstrates a much faster generation rate. The early decay observed in Ternary (H) (Fig. 2b) using the high excitation energy (550 nm), is possibly due to the fast charge motion from higher energy site to the lower energy site at the donor-acceptor interface (e.g., Grancini, Giulia et al, *Nature Mater* **12**, 29 (2013); Jakowetz, Andreas C et al, *Nature Mater* **16**, 551 (2017)).

Figure R3. Charge carrier and excited states kinetics of Ternary(H) and D1:A1 blends. a. Pump at 700 nm and probed at 1140 nm. Compared to D1:A1, there is a faster rise of charge carrier signal after 100 fs for the Ternary(H) blend, after global fitting, the rising time of Ternary (H) and D1:A1 is 67 fs and 134 fs respectively. b. Pump at 700 nm and probed at 860 nm. For the excited species (charge transfer states), after 100 fs, the excited species generated faster and

more efficient for the Ternary (H) blend. Both charge carrier and excited states kinetics results indicates that the ≈ 100 fs process contributes the efficient charge generation in the Ternary (H) blend.

As shown in the above Figure R2, we have performed the infrared kinetics measurements for the Ternary (H) blend. The difference in the kinetics between singlet (1500 nm) and charge carrier (1140 nm) signals in the initial rise can be traced after 100 fs, which is within the limit of temporal resolution of the experiment. This timescale is consistent with the observed charge separation process (Fig.2b), indicating there is efficient charge generation in the Ternary (H) blend due to the ultrafast 100 fs kinetics. Moreover, compared to the D1:A1 blend, the dynamics of charge carriers (1140 nm) demonstrates a faster rate after 100 fs (Figure R3 and Fig.S7 in Supplementary Information (SI)). The differences in the kinetics in the initial 100 fs can also be confirmed by our bias dependent transient absorption spectra (Figure 2c). Furthermore, the charge generation (1140 nm) demonstrates an almost similar feature after the initial 0.5 ps for the Ternary (H) and the D1:A1 blend (Figure R3), consistent with our transient TRTS results. Thus, the difference between D1:A1 and Ternary (H) blend in the initial 1 ps arises from the initial 100 fs process contribution (charge separation), resulting in a higher charge carrier concentration (Figure 2d). These results confirm that the initial 100 fs kinetics(charge separation) helps in the generation of the free charges in the Ternary (H) blend.

Figure 2a (Figure R2) has been added in the revised manuscript, and Figure R3 has been added in the Fig.S7 in Supplementary Information (SI). More related discussions are included on the Page 5-6 in the revised manuscript.

Reviewer's comment #3:

The authors conclude that this ultrafast decay is consistent with ballistic charge motion. Even though this might be possible, I do not understand how meaningful conclusions about ballistic charge motion can be drawn from a decay of a pump-probe signal alone, in particular if it is not clear which states are probed.

Response: We thank the reviewer for pointing out this potential confusion. As noted above, we now present the results of additional measurements (Figure R1 and Figure R2) to confirm that the 860 nm species (Figure 2b) comes from charge transfer excitons.

In the original manuscript, the sentence “such ultrafast decay independent of timescale and temperature is consistent with the ballistic motion of charge” was used. The key point of these words is that the charge separation timescale (~ 100 fs) found in the Ternary (H) blend, is consistent with the resolved ultrafast charge motion (~ 150 fs) that occurs at the donor-acceptor interface, and others have described such motion as ballistic motion (Jakowetz, Andreas C et al, *Nature*

Mater **16**, 551 (2017)). In the revised manuscript, line 97 and 98 on Page 5 have been rewritten as “This short timescale (100 fs) is consistent with the recently resolved ultrafast charge motion (150 fs) at the donor/acceptor interface”.

Reviewer’s comment #4:

The authors go on by stating that the pump-probe dynamics in Fig. 3a show “a pronounced oscillatory behavior”. Honestly, I do not see this oscillatory behavior. Even if I accept the subtraction method used by the authors to extract the “oscillation signal” in Fig. 3b, this signal does not show “clear oscillations”. (I must admit that I am very skeptical about this subtraction. I do not see much difference between the early time dynamics of the three samples. Nevertheless the data in Fig. 3b for the three samples look very different.). I see a negative peak at 45 fs and a positive peak at 100 fs in the red curve in Fig.3b. These peaks may indeed arise from an oscillatory motion with a period of 95 fs. At later times, however, this oscillatory mode is no longer seen in the data. Obviously, more and less noisy experimental data would be needed to support the claims made by the authors.

Response:

We thank the reviewer’s for thoughtful comments. The reviewer’s major concern is about the assignment of oscillation difference of three blends and related timescale of the oscillations. To address this, Figure 3a has been modified by indicating the oscillation peak (P1, P2) with an arrow (Figure R4-a). Compared to other blends, in the early time dynamics, there is a stronger oscillation kinetics in the Ternary (H) blend, as distinguished in above Figure R4-a and Figure 3a.

As the reviewer has noted, after subtraction, an oscillatory motion with a period of 95 fs is observed in the initial 200 fs (Figure R4-b and Figure 3b in the revised manuscript). At later times, the oscillation signal dephases quickly, and the weak oscillation signal is not resolved by our setup (~50 fs). To confirm that the fast dephasing of the oscillations is not due to the high noise level, 2DPS with enhanced temporal resolution of ~15 fs has been performed. In order to quantitatively analyze the oscillation signal, we monitor the quantum beat signals at two positions on the off-diagonal, as shown in Figure R4-c and Figure 3h in the revised manuscript. In the first position, pump is at 1.65 eV and probe is at 1.672 eV and in the second position, pump is at 1.644 eV and probe is at 1.634 eV. The oscillations in these two positions are in anti-phase. The oscillation signal at later times (200-300 fs) is clearly resolved, and the results also confirm fast dephasing.

Figure R4. Coherent dynamics of blends. a, Kinetic process of the three blends, pumped at 700 nm and probed at 860 nm, normalized at around 100 fs. The charge separation process occurred before 100 fs, followed by the charge generation process. The oscillation signal of Ternary (H) as indicated by the arrow. b, Corresponding oscillation signal. c, Quantum beats observed at two selected positions in the 2DPS spectra of the Ternary (H) device.

In the revised manuscript, the modified Figure 3a (Figure R4-a) and additional Figure 3h (Figure R4-c) have been added. More extensive discussion is included on the Page 12 in the revised manuscript.

Reviewer's comment #5:

Finally, the authors also present 2D photocurrent spectra. The results for a dimer are presented in Fig. 3e and S14g. I understand that the data in Fig. S14 show a single diagonal peak in the 2D spectra at an energy of about 1.64 eV (750 nm). The origin of this peak is not assigned. It seems to lie at the low-energy edge of the donor absorption. From the absorption spectra of the dimer it is unclear why excitation of this energy region should give rise to a peak in 2D spectra. The origin of the peak is not explained by the authors. I therefore understand that it is not understood. The 2D signal in Fig. S14 shows no significant changes when varying the time delay. Fig. 3e shows some (unexplained) contour line representation of this peak. Now, the peak center is shifted to an off-diagonal position – which is obviously inconsistent with the representation in Fig. S14. This discrepancy is not even mentioned in the text.

Response:

We thank the reviewer for pointing out these issues. In order to further analyze the 2D spectra, we show the evolution of the peak position as contour lines at 90% of the maximum amplitude for the D1:A1 and Ternary (H) devices. The results are shown on Figure R5-a and Figure R5-b, respectively (Figure. 3e and Figure.3f in revised manuscript). The signal rendered in colour is the phase of the nonlinear part of the periodic photocurrent, as read out from the sum and difference frequencies of the two pump and two probe pulses, with a population time separating these pulse pairs. The spectra at the zero time delay between the pump and probe shows that the main peak is shifted above the diagonal. There are different mechanisms, such as many body effects (Chemla, D.S. et al, *Nature* **411**, 549 (2001); Becker, P.C. et al. *Phys. Rev. Lett.* **60**, 2462 (1988)) that cause the shifts. Among the two effects, the upshift in the probe frequency indicates that the AC Stark effect plays the dominant role. The AC Stark effect can lower as well as increase the band gap of the material depending on the excitation spectrum (Grynberg, Gilbert, Alain Aspect, and Claude Fabre. *Introduction to quantum optics: from the semi-classical approach to quantized light*. Cambridge university press, 2010.). In our experiment, most of the excitation spectrum, as shown in Figure R5-c (Fig. 3g in the revised manuscript), is below the band gap for both the devices. In this case, the AC Stark shift increases the band gap, which is observed as a shift in the 2D peak above the diagonal. After the pump-probe overlap (i.e. beyond 15 fs), the main peak shifts to the diagonal. The peak position in both the samples (Figure R5-a and Figure R5-b, respectively) is at 1.653 eV, which corresponds to the excitonic transition.

Figure R5. A zoomed version of the 2DPS peak shifts (contour at 90% of the maximum) for a, the D1:A1 and b, Ternary (H) devices. The colour indicates the time delay between the pump and the probe. c. EQE of the two devices as a function of excitation energy, and the excitation spectrum of the laser.

The evolution of the peak position as contour lines at 90% of the maximum amplitude for the D1:A1 and for the Ternary (H) devices are shown in Figure R5-a and Figure R5-b, respectively (Figure. 3e and Figure.3f in revised manuscript).

The signal rendered in colour is the phase of the nonlinear part of the periodic photocurrent, as read out from the sum and difference frequencies of the two pump and two probe pulses, with a population time separating these pulses pairs. For the D1:A1 device (Figure R5-a and Figure 3e in the revised manuscript), the peak center is shifted in one direction to an off-diagonal position, which is not an oscillatory signal, and might come from the ultrafast excitonic relaxation process. As a comparison, a significant oscillation of the peak position is observed in the Ternary (H) device within 200 fs.

Detailed discussions are included on the Page 11-12 in the revised manuscript.

Reviewer's comment #6:

Now, in Fig. 3f and S14a, similar data are shown for the ternary blend. I must mention that it was extremely difficult to decipher these images since they were all superimposed by a bright green background. I understand, however, from S14a that the data show a similar peak as in S14b. The origin of this peak is again neither discussed nor explained. The amplitude of this peak is plotted in Fig. S14c and shows a similar dip/peak structure as in the pump-probe data in Fig. 3b. Now, however dip (75 fs) and peak (150 fs) occur at substantially different time delays. The claim of the authors that "the time scale of the oscillation is similar to that observed in the transient absorption measurements" is simply not justified. It is also fundamentally unclear to me how these peak variations can be taken as an indication of "a coherent charge transfer process". Coherences in two-dimensional spectra are observed as distinct off-diagonal peaks in the 2D map. Such peaks are neither observed nor discussed in the present data.

Response:

We thank the reviewer for pointing out this image problem, which has been resolved in the revised manuscript. As discussed in comments #5 part, in order to further analyze the 2D spectra, we show the evolution of the peak position as contour lines at 90% of the maximum amplitude for the D1:A1 and Ternary (H) devices. Results are shown in Figure R5-a and Figure R5-b, respectively (Figure. 3e and Figure 3f in the revised manuscript). The signal rendered in colour is the phase of the nonlinear part of the periodic photocurrent, as read out from the sum and difference frequencies of the two pump and two probe pulses, with a population time separating these pulse pairs. The spectra at the zero time delay between the pump and probe shows that the main peak is shifted above the diagonal. The AC Stark shift increases the band gap, which is observed as a shift in the 2D peak above the diagonal. After the pump-probe overlap (i.e. beyond 15 fs), the main peak shifts to the diagonal. The peak position in both the samples (Figure R5-a and Figure R5-b, respectively) is at 1.653 eV, which corresponds to the excitonic transition.

Figure R6. a. Quantum beats observed at two selected positions in the 2DPS spectra of the Ternary (H) device and b, the corresponding Fourier transforms. c. Out-of-plane mode. For the 203 cm⁻¹ vibration mode, mainly due to the out-of-plane vibration of 3-fluorothieno[3,4-b]thiophene(2). Out-of-plane mode (grey shadow). For the 200 cm⁻¹ vibration mode, mainly due to the out-of-plane vibration of 3-fluorothieno[3,4-b]thiophene by the C₂ and C₃ bond. The in-plane breathing mode of the benzo[1,2-b;4,5-b']dithiophene① simultaneously with the out-of-plane vibration of 3-fluorothieno[3,4-b]thiophene, aid in charge separation simultaneously during the first 200 fs.

In the similar spirit, we point out that the ternary polymer blends is rather heterogeneous. The 2DPS signal along the diagonal is slowly growing with time. The EQE is weakly dependent on energy, and does not show a distinct peak. The spectral features in the polymer blends thus resemble that of a semiconductor, with a continuum of states above the bandgap rather than a pristine molecular system. Consequently, one does not observe distinct off-diagonal peaks due to coherences. Nevertheless, the broad superimposed off-diagonal peaks do show oscillations as a function of the time delay between the pump and probe pulses. As discussed above, in 2DPS, the phase of the nonlinear part of the periodic photocurrent is read out from the sum and difference frequencies of the two pump and two probe pulses. In 2DES, one monitors the optical signal. In 2DPS we monitor the action of the four pulse sequence on the photocurrent. Thus, compared to 2DES, 2DPS directly probes the intermediate states that contribute to the photocurrent.

Thus, in order to quantitatively analyze the oscillation frequency, we monitor the quantum beat signals at two positions on the off-diagonal as shown in Figure R6-a and Figure. 3h in the revised manuscript. In the first position, pump is at 1.65 eV and probe is at 1.672 eV and in the second position, pump is at 1.644 eV and probe is at 1.634 eV. The oscillations in these two positions are in anti-phase (Figure R6-a and Figure. 3h in the revised manuscript). The Fourier transforms show significant vibrational contribution to the photocurrent at frequencies below

500 cm^{-1} . The main peak in the spectra is around 203 cm^{-1} which is different from the main peak in the TA results. This result can be rationalized based on the fact that in photocurrent detected 2D, the results are susceptible to the EQE of the photocurrent, while in the TA we only measure the optical response. Quantum calculation results further indicates that the low vibrational mode ($\sim 203 \text{ cm}^{-1}$) can be mainly attributed to the out-of-plane vibration of the 3-fluorothieno[3,4-b]thiophene donor unit in D1 (Figure R6-c and Figure 4c in the revised manuscript). Such out-of-plane conformer would hinder close main-chain stacking and influence the local configuration of the A1 moieties near the D1 backbone (Graham, K.R. et al, *J. Am. Chem. Soc.* **136**, 9608 (2014)), facilitating the intermolecular charge transport between D1 and A1. The results clearly indicate a coherent charge transfer process within 200 fs of the excitation, that has a significant contribution to the photocurrent in the Ternary (H) device.

Detailed discussions are included on the Page 11-12 in the revised manuscript.

Reviewer's summary comment #7:

I therefore conclude the data reported in the manuscript do not support the claim expressed in the title of this paper. Publication of those data is clearly premature and cannot be recommended. Additional experimental evidence is needed before claims about coherence in charge separation processes can be made. Such claims must be based on a thorough data analysis – which is not presented in the current work. I therefore do not recommend to publish this work in Nature Communications.

Response: We greatly appreciate the reviewer's thoughtful review of the manuscript and the constructive suggestions. The reviewer's main concern on the assignments of the charge transfer excitons spectral feature and coherence discussions might be caused by the missing of some key experimental data and detailed 2DPS interpretation in the original manuscript, which we hope we have now provided in responding to reviewer's comments. Reviewer's critical comments have pushed us to carry out several additional experimental measurements, in particular TA in the near infrared and more detailed 2DPS analysis, which provide more solid experimental evidences for our claims. We have addressed all the comments/concerns raised by the reviewer. Following the reviewer's suggestion, we have substantially modified the manuscript in the discussion section on Page 5-6 and on Page 11-12 of the revised manuscript. In addition, we have polished the language in the revised manuscript. With the substantial new experimental data and additional discussions in addressing the reviewer's comments, as well as the much-modified presentation, the revised

manuscript is significantly improved. Hopefully, the reviewer will now find the revised manuscript suitable for publication in Nature Communications.

Reviewer's comment #8:

Finally, I want to remark that the discussion of “coherence” that is given in this paper is rather superficial since no distinction between electronic, vibronic or vibrational coherences is made. A more in-depth explanation of what the authors mean by “coherence” should be given.

Response: We have clearly pointed out that the coherences are of vibronic nature in the revised manuscript.

For the more theoretical nuances of these different forms of coherence, we refer to references (e.g., Chenu, A et al, *Annu. Rev. Phys. Chem.* **66**, 69 (2015); Scholes, Gregory D, et al, *Nature* **543**, 647 (2017)). To our knowledge, electronic coherence helps in charge or energy transfer process only proposed in some efficient biological systems (e.g., Ma, F et al, *Nat. Commun.* **10**, 933 (2019); Thyryhaug, E et al, *Nat. Chem* **10**, 780 (2018)). The coherence time found in our study, completed in the first 200 fs, which is similar to the resolved electronic coherence (e.g., Ma, F et al, *Nat. Commun.* **10**, 933 (2019); Thyryhaug, E et al, *Nat. Chem* **10**, 780 (2018)) and significantly different from the resolved vibrational coherence, which observed in the polymer semiconductor (e.g., De Sio, A et al. *Nat. Commun* **7** 13742 (2016); Falke, S.M., et al. *Science* **344**, 1001 (2014)). Though we are not able to exclude the influence of purely electronic or vibrational coherence in the charge generation by experimental results, our joint experimental and quantum calculations suggest that coupled electronic and vibrational coherence, that is vibronic coherence, is the more plausible candidate.

Reviewer's comment #9:

2D-photocurrent spectra of organic materials have been studied by Bittner et al (and others). References to this work should be added.

Response: We appreciate the reviewer for reminding us of these studies. As reviewer noted, Eric R. Bittner and others have also used 2D coherent spectroscopy (Bittner, E. R et al, *Chem. Phys.* **481**, 281 (2016)) and quantum-dynamical analysis (e.g., Bittner, E. R et al, *Phys. Rev. Lett.* **100**, 107402 (2008); Bittner, E.R et al, *Nat. Commun.* **5**, 3119 (2014)) to study the coherent exciton dissociation at polymer heterojunctions. Related papers have been cited in the revised manuscript.

Response to the Comments of Reviewer #3:

Reviewer's Summary Comments:

*Optoelectronic properties and ultrafast energy and charge carrier dynamics in several bulk-heterojunction organic semiconductor devices have been examined with a number of complementary state-of-the-arts electronic and vibrational spectroscopies to probe appearance of electron-vibrational (vibronic) dynamics. The authors conclude that ultrafast transformation of tightly-bound photoexcited exciton into the charge-separated state on ~200fs timescale is facilitated by a coherent vibronic dynamics associated with the breathing mode on the polymer at ~365 cm⁻¹. Such observations add valuable detailed experimental observation into decades-long arguments on how coherent evolution of electronic and vibrational degrees of freedom affect dynamics of carriers and may have consequences for future organic light-harvesting and lighting applications. Overall, the experimental results are convincing, well presented and the manuscript is well written. Consequently, the article potentially may be published in *Nature Communications*.*

*However, the main conclusions of the paper are not strongly supported by the results. Thus, I do not recommend this paper for publication in *Nature Communications* in its present form. I hope that the authors will be able to address these issues in the revision.*

Response: We thank the reviewer for the encouraging comments. We have addressed the shortcomings pointed out by the reviewer in the revised manuscript and believe that the evidence we have provided now adequately support our claim. We also believe that the measurements we have presented give the most direct evidence of the role played by vibronic coherences in the generation of photocurrent in polymer solar cells.

Reviewer's comment #1:

Main critique: The main conclusion of this work is that "The vibrational mode found here can be used to inspire the design of new photovoltaic materials". This vibration is identified as a breathing mode of D1 donor compound. To me, the central question remains unanswered: why does this vibration promote charge separation in the tertiary H device (D1:D2:A1) while it is not affecting the binary device (D1:A1) performance? What is so special in mixing in D2 compound which does not have such vibrations and has higher lying excitonic states (that seemingly should not participate in charge separation)?

Response: We thank the reviewer's thoughtful comments. The reviewer's major concern is about the contribution of D2 in the observed coherence process of the Ternary (H) system. This is an insightful question. In the revised manuscript, the observed coherence is interpreted by the transition from the primary excited states (EX) to charge separated states (CS), through resonance with the low frequency vibrational mode. The coherence effects results from such strong resonance interactions are robust and decisive in their roles for function, that means these states are little perturbed by environment, like disorder and fluctuating interaction and more. The resonance contribution also means that CS states can be maintained in phase when the system is subject to strong random fluctuations. To achieve such resonance contribution, the energy gap between the initial (EX) and product (CS) states is a key parameter, since the energy gap fluctuations affect resonance (Zhang, Yuqi, et al. *Proc. Natl. Acad. Sci.* **111**,10049 (2014)). Due to the contribution of D2, (intermolecular interactions between D1 and D2 and acceptor), the local molecular configuration of D1 in the Ternary (H) blend may be different from D1:A1 blend, and this could result in a different EX and CS states (and also a different energy gap value). Such energy gap fluctuations may not be resonant with the low frequency breathing mode, and thus only weak coherence or decoherence occurs. This might be one reason why significant coherence process is not observed in D1:A1 system.

We have further analyzed the 2D spectra to quantify the oscillation frequency. We monitor the quantum beat signals at two positions on the off-diagonal as shown in Figure R6-a and Figure. 3h in the revised manuscript. In the first position, pump is at 1.65 eV and probe is at 1.672 eV and in the second position, pump is at 1.644 eV and probe is at 1.634 eV. The oscillations in these two positions are in anti-phase (Figure R1-a and Figure. 3h in the revised manuscript). The Fourier transforms show significant vibrational contribution to the photocurrent at frequencies below 400 cm^{-1} (Figure R1-b and Figure 3i in the revised manuscript). The main peak in the spectra is around 200 cm^{-1} which is different from the main peak in the TA results. Quantum calculation results further indicates that the low vibrational mode ($\sim 200 \text{ cm}^{-1}$) can be mainly attributed to the out-of-plane vibration of the 3-fluorothieno[3,4-b]thiophene donor unit in D1 (Figure R1-c and Figure 4c in the revised manuscript). Such out-of-plane conformer would hinder close main-chain stacking and influence the local configuration of the A1 moieties near the D1 backbone (Graham, K.R. et al, *J. Am. Chem. Soc.* **136**, 9608-9618 (2014)), which facilitates the intermolecular charge transport between D1 and A1. This also means that the A1 docks with a specific part of D1(the in-plane breathing or out-of-plane vibration or both of these two configuration), and the intermolecular interactions between D1 and D2 definitely

affect such local molecular configuration. As a result, the charge transfer between D1 and A1 is different between D1:A1 and Ternary (H) systems (as confirmed by the interpretation of ultrafast charge separation part in the manuscript). Furthermore, such local molecular configuration could influence the electronic coupling between the EX and CS states (e.g., Yi, Yuanping et al, *J. Am. Chem. Soc.* **131**, 15777 (2009); Wang, Tonghui, et al. *Adv. Funct. Mater.* **28**, 1705868 (2018)), which facilitates the transition between these two states. This might be another reason why such vibronic coherences are not observed in the D1:A1 blend. Unfortunately, the precise information on such local intrachain conformations or interchain configurations might be accessed only by complex theoretical calculation.

Figure R1. a. Quantum beats observed at two selected positions in the 2DPS spectra of the Ternary (H) device and b, the corresponding Fourier transforms. c. Out-of-plane mode (grey shadow). For the 200 cm⁻¹ vibration mode, mainly due to the out-of-plane vibration of 3-fluorothieno[3,4-b]thiophene by the C₂ and C₃ bond. The in-plane breathing mode of the benzo[1,2-b;4,5-b']dithiophene ① simultaneously with the out-of-plane vibration of 3-fluorothieno[3,4-b]thiophene, aid in charge separation simultaneously during the first 200 fs.

Reviewer's comment #2:

The authors convincingly “demonstrate that the differences in morphology, interface area and disorder cannot explain our observed differences among the three blends.” Why does then 7:3:10 mixture activate this breathing mode compared to 9:1:10 and 10:0:10 mixtures for D1:D2:A1 blend?

Response: We thank the reviewer's thoughtful comments. The reviewer's major concern is about the reason why the observed coherence process only is significant in the Ternary (H) system with the specific D1:D2 ratio. As explained in **comment #1**, the coherence effects results from strong resonance interactions between EX and CS states, and the energy gap fluctuations affect such resonance. Due to the contribution of D2, (intermolecular interactions between D1 and D2 et al.), the local molecular configuration of D1 in the Ternary (H) blend will be

different from D1:A1 blend, and this could result in different EX and CS states (also a different energy gap value). Different D1:D2 ratio also means the degree of intermolecular interaction between D1 and D2 is different, which results in different energy gap between EX and CS states. In other D1:D2 ratio blends (like 9:1), the resulting energy gap fluctuations may not form resonance with the low frequency breathing mode, and only weak coherence or decoherence occurs. This speculation is consistent with the observed oscillation signal (Figure 3b in the revised manuscript) in that, compared to Ternary (H), a much weaker oscillation amplitude can be observed in the 9:1:10 and 10:0:10 mixtures for D1:D2:A1 blend. The local molecular configuration could influence the electronic coupling between the EX and CS states. Coherence relates to phase relations among the constituents in the superposition of quantum states, and such phase relationships among quantities (quantum amplitude) are retained long enough to have a functional relevance to the underlying process (EX-CS). Much smaller electronic coupling (localized electronic states) or much larger electronic coupling (strongly delocalized electronic states) results in incoherent charge or energy transport. As confirmed from the transient results (Supplementary Fig.S18 in Supplementary Information (SI)), the competing excited states energy transfer (EES) process is significant in the 9:1:10 and 10:0:10 mixtures for D1:D2:A1 blend, which means the electronic coupling no longer meet above conditions, and as a result, there is no significant coherence in these blends.

Reviewer's suggestions #1:

Abstract: "The vibrational mode found here can be used to inspire the design of new photovoltaic materials." I believe that the authors would like to say that the presented efficient ultrafast charge separation process assisted by specific coherent vibrational motion, can be used to inspire....

Response: We thank the reviewer's valuable suggestions. The improved description "This efficient ultrafast charge separation process coupled with specific low frequency coherent vibrational motion, can be used to inspire the design of new photovoltaic materials with high device performance" has been used in the lines 25-27 in the revised manuscript.

Reviewer's suggestions #2:

P. 5: "After 0.5 picoseconds (ps), the kinetic curve represents the deactivation of the lowest-level excited state." Pls clarify if you mean the lowest exciton is D1 state.

Response: We thank the reviewer's careful reading and pointing out this description issue. An improved description sentence "After 0.5 picoseconds (ps), the kinetic curve represents the deactivation of the lowest-level excited state of

D1” has been used in Fig.S9 in Supplementary Information (SI) in the revised manuscript.

Reviewer’s suggestions #3:

P. 7: “resulting charge generation process is entropy controlled”. Clarify, why?

Response:

Figure R2. a. Temperature dependent photovoltage evolution for three blend diodes. b. Transient THz photoconductivity kinetics ($\lambda_{\text{exc}}=400$ nm, $I_{\text{exc}} = 4.5 \times 10^{12}$ ph/cm²).

At the donor-acceptor interface, excitons (single species) dissociate to separated charges (two species), this means that the number of electronic states available to electrons increase as the separation increase, thus, entropy might contribute to the charge generation process. Theoretical analysis found that entropy increased with the separation distance (e.g., Clarke, Tracey M. et al, *Chem. Rev.* **110**, 6736 (2010); Gregg, Brian A, *J. Phys. Chem. Lett.* **2**, 3013 (2011)). Previous study (Gao, Feng, et al, *Phys. Rev. Lett.* **114**, 128701 (2015) demonstrates that temperature independent charge separation ($V_{\text{oc}}-T$ plots) can be interpreted by the electron-hole separation distance, which is related to the entropy contribution. In the manuscript, we do the temperature dependent V_{oc} test (Figure R2-a and Figure 2e in the revised manuscript). Different from reference devices, the V_{oc} of Ternary (H) device keeps increasing with decreasing temperature (to 110K). This means that the charge separation process still is efficient below 110K and therefore is temperature independent above this temperature, due to the entropy contribution that cancels the Coulombic attraction. Furthermore, this temperature independent charge carrier generation, as deduced from $V_{\text{oc}}-T$ plots, means the photogenerated carrier typically is not fully relaxed, a process which normally occurs on the picosecond-nanoseconds timescale (e.g., Melianas, A. et al, *Nat. Commun.* **6**, 8778 (2015); Van Eersel, H. et al, *Adv. Funct. Mater.* **22**, 2700-2708 (2012)), such timescale close to our transient THz test (Figure R2-b and Figure 2d in the revised manuscript).

Reviewer's suggestions #4:

P11: "the energy gap between the initial EX (emission peak ~800 nm) and the CS 229 states (emission peak ~780 nm), is resonant with the breathing mode (~360 cm⁻¹)." Unclear. This is significant. Why should the resonance between fully relaxed excitonic state and the CS state be important? I do not see any conclusive evidence to make this statement (except the numbers are lining up ok). It seems to me that the transition to CS occurs dynamically before equilibration. And this also should be present in D1:A1 mixture but does not play a significant role.

Response: This is an insightful question. Normally, the transition to CS is believed to occur dynamically before equilibration. A recent study found that the charge generation process can be efficient from the totally relaxed charge transfer excitons (Vandewal, K. et al, *Nat. Mater.* **13**, 63 (2014)). The study found that there is ultrafast charge generation (~45 fs) from the totally relaxed singlet (Grancini, G. et al, *Nat. Mater.* **12**, 29 (2013)). These results mean that the transition to CS could possibly occur after equilibration. From the experimental results (Figure 4b in the manuscript), we found the energy gap (~ 40 meV) between the initial EX and CS states is close to the breathing mode (360 cm⁻¹, ~ 44 meV); thus the breathing mode could possible accomplish resonance between these two states. Coherence effects that result from strong resonance interactions are robust and decisive in their roles for function, which means that these states are little perturbed by the environment, like disorder and fluctuating interaction etc. The resonance contribution also means that CS states can be maintained in phase when the system is subject to strong random fluctuations, and this might be one reason why coherence is observed in this specific Ternary system. Furthermore, as the transition from EX to CS is an entropy increasing process (as explained in **response #3**), the low frequency vibrational modes might be responsible for the charge separation and are likely populated by dissipation of excess bond stretching energy created during photoexcitation.

Reviewer's suggestions #5:

Fig. 1. Panel b) Transient absorption curves are too tiny... these may deserve the entire panel. Panel c) colors in the cartoon for D1 and D2 are too similar to be easily distinguishable.

Response: We thank the reviewer for these valuable suggestions. Following the reviewer's suggestion, we have now included the transient absorption curves in another figure (Figure R3-a below and Figure 1b in the revised manuscript). Further, an improved cartoon has been used in the revised manuscript (Figure R3-

b below and Figure 1c in the revised manuscript), relevant parameters can be clearly distinguished.

a

b

Figure R3 a. UV-Vis absorption for PCDTBT (D2, blue solid) and PTB7-th (D1, black solid); Transient absorption signal (~ 10 ps) from the investigated pristine and blend films, pumped at 550nm. b. Simple cartoon illustration of the charge photo-generation at a heterojunction. Light absorption generates excitons (EX) at the interface sites (1), then undergo coherent charge separation (CCS) into loosely bound electron-hole pairs ((2), charge separated states (CS)). The electron and hole separate further and form free charges (3).

Reviewer's suggestions #6:

Fig. 3f: appears as a green square on the screen. Pls check the original version.

Response: We thank the reviewer for pointing out this image problem, and the original figure (Figure R4 below and Figure 3f in the revised manuscript) can be found in the revised manuscript.

Figure R4. A zoomed version of the 2DPS peak shifts (contour at 90% of the maximum) for the Ternary (H) devices. The colour indicates the time delay between the pump and the probe.

Reviewer's suggestions #7:

General comments for all Figures: the fonts are small and the lines are too thin to be easily readable. This should be fixed for production version.

Response: We thank the reviewer for pointing out these figure issues. We have improved all figures in the revised manuscript according to the reviewer's suggestion. The improved figures meet the requirements for the production version.

Reviewer's suggestions #8:

On the screen presented calculated vibrational modes are not well seen. The pictures look like the molecular views from different angles.

Response: We thank the reviewer for pointing out this image issue. An improved image has been used in the revised manuscript. To clearly visualize such vibration, the related video file is also used as "MP4 files".

Reviewers' comments:

Reviewer #1 (Remarks to the Author):

The authors have presented additional experiments that support the initial interpretation of their data. Despite these advances and improvements, I still find the manuscript too speculative to merit publication in a high profile journal in its present form.

How do the authors know that the transient oscillatory features that they observe are "responsible for the ultrafast charge separation"?

Why are the transient signals that are seen in Fig. R4b so very similar in shape (red curve in (b) and blue curve in (c)), but their oscillation periods so very different? Why should is one of the modes "responsible for charge separation" but the other one not? [This problem is a significant one: Reviewer#3 summarizes: "The authors conclude that ... charge separation ... is facilitated by ... the breathing mode on t the polymer at 365 cm^{-1} .". Now the authors see a 200 cm^{-1} (but not 365 cm^{-1} mode) in the transient photocurrent data. It is not clear to me how these conflicting results can be reconciled.]

Why can an oscillation amplitude of 5% - 10% of the total signal in Fig. R4a be used to support the claim "this oscillatory feature is responsible for ultrafast charge separation". It seems to me that - at most - there is a certain small contribution of coherent transients to the overall charge transfer yield.

Why do the spectra in Fig. 1b not show clear transient absorption resonances that can safely be assigned to "charge-transfer states", as claimed by the authors?

Why do the transient pump-probe data detected for probe wavelengths near 1500 nm show a clear decay in amplitude - assigned to an exciton dissociation process - even though the spectra in this range do not show a clear exciton resonance?

Why do the authors represent the 2D spectra in Fig. R5 as contour lines - without showing the spectral line shape of the 2D amplitude spectra? How can the authors claim that the 2D spectra reflect an Optical Stark Effect? AC Stark shifts usually result in distinct dispersive line shapes in pump-probe or 2D spectra. These dispersive line shapes are not shown, and therefore, it is not clear to me how the authors can assign their spectra to AC Stark shifts.

The authors assign their coherences to "vibronic coherences". What is the firm basis for this assignment? Vibronic coherences imply correlations between coherences in the electronic and vibrational degrees of freedom. How can these correlations be deduced from the present data? I can only recommend this paper for publication in Nature Communications after all these questions have been thoroughly and convincingly addressed.

Reviewer #3 (Remarks to the Author):

The authors have provided an adequate and exhaustive response to the referee's critique. The MS was modified accordingly. Now I believe that the article can be accepted for publication.

Response to Reviewers' Comments on NCOMMS-19-18303-A

We thank the reviewers for their constructive comments and careful reviews of our manuscript, which have helped us to further improve our manuscript. We note that while Referee 3 accepts our revision in full, there are remaining and repeated questions from Referee 1. In the past weeks, we have modified the presentation in our manuscript to address all these comments/questions. In the following, we present our point-to-point responses to all the reviewers' comments and indicate corresponding changes made in the revised manuscript. The major changes have been highlighted by colour in the "revised manuscript with marks". We believe that this revised manuscript is much improved from the original version, and could now be accepted for publication in *Nature Communications*.

Response to Reviewer #1's comments:

Reviewer #1 (Remarks to the Author):

The authors have presented additional experiments that support the initial interpretation of their data. Despite these advances and improvements, I still find the manuscript too speculative to merit publication in a high profile journal in its present form.

Reviewer's comment #1:

How do the authors know that the transient oscillatory features that they observe are "responsible for the ultrafast charge separation"?

Response: We thank the reviewer's thoughtful comments. The initial rise in the kinetics of the Ternary (H) blend shows a pronounced oscillatory behavior (Figure R1a and Figure 3a in the revised manuscript), with an amplitude that is significantly higher than in the other two blends (Figure 3b in the revised manuscript). After this initial rise (~100 fs) kinetics, there is a faster charge transfer excitons generation in the Ternary (H) blend, which was confirmed by the higher amplitude in the later 1 ps timescale (Figure R1a). Furthermore, a significantly increased charge carrier generation rate after the initial 100 fs was observed in the Ternary(H) blend (Figure R1b and Figure S7b in the Supplementary Information (SI)). These results indicate that the initial 100 fs kinetics (which demonstrates oscillatory behavior) is responsible for the efficient charge generation in the Ternary (H) blend. We have further analyzed the Raman activity corresponding to the in-plane breathing mode. Compared to the neutral, the positively charged CT conformer demonstrates a significant enhancement of the Raman activity (Figure R1c and Figure 3d in the revised manuscript). This

result indicates a breathing mode is in resonance with the ultrafast charge transfer process, which results in efficient CS states formation, consistent with our experimental results. The ultrafast charge separation results in the efficient charge generation in the initial 1 ps, which is explained in the line 105-115 part in the revised manuscript. The same operating timescale (in the first 100 fs) of the transient oscillatory and ultrafast charge separation; Furthermore, both the transient oscillatory and ultrafast charge separation contributes to the efficient charge generation at later timescale (~ 1 ps). These results suggest that the initial transient oscillatory features are coupled with the ultrafast charge separation. We also note this in devices, where the existence of generated charge carriers in the hundreds of femtoseconds under reverse bias (Figure 2c in the revised manuscript), and the 2DPS signal (Figure 3h in the revised manuscript) reveal ultrafast charge generation demonstrates oscillatory behavior within 200 fs. The transient results from both film and device are consistent with each other, and suggest the initial transient oscillatory features are coupled with the ultrafast charge separation.

Figure R1. a, Excited states kinetics of Ternary(H) and D1:A1 blends. Pump at 700 nm and probed at 860 nm. b, Charge carrier kinetics of Ternary(H) and D1:A1 blends. Pump at 700 nm and probed at 1140 nm. c, Raman spectra calculated at the B3LYP/6-31+G(d) level of theory for the neutral and charged CT conformer.

Detailed discussions are included in the line 198-207 and line 213-218 in the revised manuscript.

Reviewer's comment #2:

Why are the transient signals that are seen in Fig. R4b so very similar in shape (red curve in (b) and blue curve in (c)), but their oscillation periods so very different?

Response: We thank the reviewer's thoughtful comments. As the reviewer pointed out, the resolved oscillation period is different in transient absorption and 2D photocurrent spectra. The Fourier transforms show significant vibrational contribution to the photocurrent at frequencies below 500 cm^{-1} (Figure R2a and Figure 3h in the revised manuscript). The main peak in the spectra is around 200 cm^{-1} which is different from the main peak in the TA results (Figure R2b and Figure 3b in the revised manuscript). This result can be rationalized based on the fact that in photocurrent detected 2D, the results are susceptible to the EQE of the photocurrent, while in the TA we only measure the optical response. Note that the interval of oscillation frequencies (Figure R2c and Figure 3i in the revised manuscript) includes the vibrational mode that is resolved by TA. Similar to the TA results, the oscillations dephase quickly ($\sim 200\text{ fs}$). Both vibrational modes ($\sim 365\text{ cm}^{-1}$ and 200 cm^{-1}) aid in charge separation simultaneously and persist in the first 200 fs. Our results show that the low frequency vibrational modes have greater contribution to the coherent photocurrent generation. The low vibrational mode ($\sim 200\text{ cm}^{-1}$) can be mainly attributed to the out-of-plane vibration of the 3-fluorothiopheno[3,4-b]thiophene donor unit in D1 (Figure R2d and Figure 4c in the revised manuscript). The in-plane breathing mode of the benzo[1,2-b;4,5-b']dithiophene^① simultaneously with the out-of-plane vibration of 3-fluorothiopheno[3,4-b]thiophene (grey shadow), aid in charge separation simultaneously during the first 200 fs.

Figure R2. Coherent dynamics of blends. a, Quantum beats observed at two selected positions in the 2DPS spectra of the Ternary (H) device. b, Oscillation signal of the three blends. Detected by transient absorption, pumped at 700 nm and probed at 860 nm. c, Fourier transforms of quantum beats, in the 2DPS spectra of the Ternary (H) device. d, Out-of-plane mode (grey shadow). The in-plane breathing mode of the benzo[1,2-b;4,5-b']dithiophene^① simultaneously with the out-of-plane vibration of 3-fluorothieno[3,4-b]thiophene, aid in charge separation simultaneously during the first 200 fs.

Detailed discussions are included in the line 273-279 in the revised manuscript.

Reviewer's comment #3:

Why should is one of the modes “responsible for charge separation” but the other one not? [This problem is a significant one: Reviewer#3 summarizes: “The authors conclude that ... charge separation ... is facilitated by ... the breathing mode on t the polymer at 365 cm⁻¹.”. Now the authors see a 200 cm⁻¹ (but not 365 cm⁻¹ mode) in the transient photocurrent data. It is not clear to me how these conflicting results can be reconciled.]

Response: We thank the reviewer's thoughtful comments. This main reservation of the reviewer, which was previously mentioned by reviewer 3, has already been addressed in the manuscript. Here, we would like to point out that the two modes can have different contributions in the coherent optical signal and the photocurrent signal. Both the pump-probe and the photocurrent 2D show contributions from the vibrational modes below 500 cm⁻¹, which is expected. The pump-probe results show which modes participate in the charge separation. However, this does not necessarily mean that all the modes contribute equally to the photocurrent. The final photocurrent quantum yields can differ for the different modes. Our results show that the modes around 200 cm⁻¹ have larger contribution to the photocurrent.

Reviewer's comment #4:

Why can an oscillation amplitude of 5% - 10% of the total signal in Fig. R4a be used to support the claim “this oscillatory feature is responsible for ultrafast charge separation”. It seems to me that – at most – there is a certain small contribution of coherent transients to the overall charge transfer yield.

Response: We thank the reviewer's thoughtful comments. The reviewer's major concern is about the contribution of the resolved coherent transients to the overall charge transfer yield. This is an insightful question. In the manuscript, the absolute coherence contribution for the overall charge transfer yield was not discussed. For the Figure R4a (Figure 3a in the revised manuscript), the spectra were normalized around 100 fs, and the oscillation amplitude of 5%-10% is only small part of the total oscillation (the total oscillation persistent in the first 200 fs). To our knowledge, it is really hard to precisely determine the absolute contribution of coherence in these transient studies. One strategy is evaluating the charge transfer rate and oscillatory signal, exhibiting enhanced signal intensity of

physical process and vibronic character (Wang, L. et al. *Nat. Chem.* **9**, 219(2017); Scholes, G.D. et al. *Nature* **543**, 647(2017)).

In similar spirit, we can further roughly evaluate the coherence contribution in the Ternary (H) blend. As explained in **Comment#1**, the initial transient oscillatory features (~ 100 fs) coupled with the ultrafast charge separation, and responsible for the charge generation which occurred on the later timescale (~ 1 ps). Charge transfer excitons (Figure R4a and Figure 3a in the revised manuscript) and charge carrier kinetics (Figure R4b and Figure S7b in the Supplementary Information (SI)) results suggest the enhanced charge transfer excitons and charge carrier generation due to the increased generation rate, which is coming from the initial 100 fs transient oscillatory contribution. As a result, almost 75% enhancement of the charge transfer excitons was observed in the Ternary (H) blend compared to D1:A1 blend in the initial 1 ps. The enhancement of charged species in the first 1 ps of the Ternary (H) blend is consistent with the transient THz results (Figure R4c and Figure 2d in the revised manuscript), almost 80% enhancement of the charge carriers was observed in the Ternary (H) blend compared to D1:A1 blend in the initial 1 ps. The calculated mobility of ~ 0.55 cm^2/Vs estimated for the Ternary (H) blend is higher than in the D1:A1 blend (0.3 cm^2/Vs). Furthermore, this also means an ultrafast charge generation preceding the time scale that TRTS can access (<100 fs), resulting in the increased photoconductivity in the Ternary (H) blend in the initial 1 ps. These results suggest that the ultrafast ~ 100 fs transient oscillatory significantly contributes to the final enhancement of the charge generation in the Ternary (H) blend.

Figure R4. a, Excited states kinetics of Ternary(H) and D1:A1 blends. Pump at 700 nm and probed at 860 nm. b, Charge carrier kinetics of Ternary(H) and D1:A1 blends. Pump at 700 nm and probed at 1140 nm. c, Transient THz photoconductivity kinetics ($\lambda_{\text{exc}}=400$ nm, $I_{\text{exc}} = 4.5 \times 10^{12}$ ph/cm 2).

Reviewer's comment #5:

Why do the spectra in Fig. 1b not show clear transient absorption resonances that can safely be assigned to "charge-transfer states", as claimed by the authors?

Response: This is a technical comment. We agree with the reviewer that the charge transfer states will show transient absorption resonances. As explained in the line 81-87 in the revised manuscript, the 860 nm PIA band increases

simultaneously with the singlet exciton dissociation in the initial picoseconds (Figure 2a in the revised manuscript), which helps us to safely assign the charge transfer states. The Figure 1b is the transient absorption signal at 10 ps timescale, the charge transfer process is almost completed (as can be seen from Figure R4a) and will demonstrate a constant feature instead of resonances. A resonance feature of the charge transfer excitons will occur on the initial timescale (< 10 ps), as can be seen from the Figure S3 in the Supplementary Information (SI).

Reviewer's comment #6:

Why do the transient pump-probe data detected for probe wavelengths near 1500 nm show a clear decay in amplitude – assigned to an exciton dissociation process – even though the spectra in this range do not show a clear exciton resonance?

Response: This is a technical comment. As explained in the Figure S4 in the Supplementary Information (SI), the PIA band peaking at 1500 nm assigned to singlet states, that shows a decay that matches both CT and charge carrier PIA rise. This assignment consistent with previous reports (Tamai, Yasunari, et al. *ACS nano* **11**,12473(2017); Shivanna, R., *Energy. Environ. Sci*, **7**, 435 (2014)). Due to our setup limitations, the signal-to- noise is not very well resolved. While the kinetics curves (Figure 2a in the revised manuscript) with improved signal-to- noise, demonstrates a clear decay kinetics in the initial picoseconds, suggest the signal come from excited singlet exciton.

Reviewer's comment #7:

Why do the authors represent the 2D spectra in Fig. R5 as contour lines – without showing the spectral line shape of the 2D amplitude spectra? How can the authors claim that the 2D spectra reflect an Optical Stark Effect? AC Stark shifts usually result in distinct dispersive line shapes in pump-probe or 2D spectra. These dispersive line shapes are not shown, and therefore, it is not clear to me how the authors can assign their spectra to AC Stark shifts.

Response: The 2D amplitude spectra are given in the supplementary information Fig. S17. The contour lines have been used to highlight the most relevant information from the overall spectra. Readers interested in complete amplitude spectra can find them in the supplementary information.

The spectra, in general, have contributions from three types of signals, the ground state bleach (GSB), stimulated emission (ESA) and excited state absorption (ESA). As opposed to the traditional type of 2D spectra where one detects coherent four-wave mixed signal, photocurrent 2D spectra have two types of ESA signals, which are often called as ESA I and ESA II. The Feynman diagrams depicting the different light-matter interaction pathways (Figure R5 and Figure S18 in the revised Supplementary Information). The diagrams show the evolution of the density matrix of the system when perturbed by the light. The ESA I and ESA II differ in the final state that is reached after the four interaction. In ESA I,

the system is finally in the first excited state $|1\rangle$, while in ESA II it is in the high lying excited state $|2\rangle$. Each of the signals have a sign associated to it, which is given by $(-1)^{n-1}$, where n is the number of interactions from the bra (or the ket) side of the density matrix (Damtie, F.A., et al. *Phys. Rev. A* **96**, 053830 (2017)). According to the diagrams, the GSB, SE and ESA I have the positive sign while the ESA II has the negative sign. In most of the solar cells, the excitation of high lying states does not yield higher photocurrent yield compared to the excitation at the band-edge because the excess energy is dissipated as heat due rapid thermal relaxation. Consequently, the ESA I and ESA II signals cancel each other, such that the 2D photocurrent signal effectively comes from the GSB and SE pathways. Spectra from these two pathways have only absorptive lineshapes with only positive peaks that are similar to the ones we observe in our 2D photocurrent measurements. There are special cases, such as photocurrent from quantum dot based devices, where the photocurrent quantum yield from the high lying states can be higher due to the generation of multiple excitons from the excess energy. In such cases the lineshapes can be dispersive having positive and negative features (Karki, K.J. et al. *Nat. Commun.* **5**, 5869 (2014)). Consequently, the spectral shifts due to AC Stark effect does not produce dispersive lineshapes. As the AC Stark effect shifts the GSB and SE contributions to above the diagonal when the excitation photons have energies less than the band-gap, we have assigned the shifts observed in our measurements to this effect. Other possibilities have also been mentioned in the manuscript. However, they don't explain our observations.

Figure. R5. The Feynman diagrams depicting the different light-matter interaction pathways.

Detailed discussions are included in the Figure S18 in the revised Supplementary Information (SI).

Reviewer's comment #8:

The authors assign their coherences to “vibronic coherences”. What is the firm basis for this assignment? Vibronic coherences imply correlations between coherences in the electronic and vibrational degrees of freedom. How can these correlations be deduced from the present data?

Response: We thank the reviewer’s thoughtful comments. In the manuscript, we demonstrate that the resolved beating frequencies (365cm^{-1}) resonant with one Raman mode of D1, this result suggest the vibrational coherence contribute to this resonance. While a purely vibrational mechanism can be ruled out if we account for the observation that such coherence only is significant in the Ternary (H) blend, which was confirmed by both transient absorption and 2D photocurrent results. If it is vibrational coherence, the binary blend will also demonstrate such quantum beats features, since the resolved 365cm^{-1} mode come from D1. On the other hand, the results can be explained by the vibronic coherence through the resonance contribution. Vibronic coherences arise from the quantum superpositions of a vibronically coupled system (the excited states have mixed electronic–vibrational character), this means that the vibrational and electronic motions are not independent of one another. In the manuscript, the observed coherence is interpreted by the transition from the primary excited states (EX) to charge separated states (CS), through resonance with the low frequency vibrational mode. The resonance contribution means that CS states can be maintained in phase when the system is subject to strong random fluctuations. To achieve such resonance contribution, the energy gap between the initial (EX) and product (CS) states is a key parameter, since the energy gap fluctuations affect resonance (Zhang, Yuqi, et al. *Proc. Natl. Acad. Sci.* **111**,10049 (2014)). Due to the contribution of D2, (intermolecular interactions between D1 and D2 and acceptor), the local molecular configuration of D1 in the Ternary (H) blend may be different from D1:A1 blend, resulting in different energy gap between EX and CS states. Furthermore, the resolved vibrational mode ($\sim 200\text{cm}^{-1}$) can be mainly attributed to the out-of-plane vibration (Figure R2d and Figure 4c in the revised manuscript); such an out-of-plane conformer would hinder close main-chain stacking and influence the local configuration of the A1 moieties near the D1 backbone (Graham, K.R. et al, *J. Am. Chem. Soc.* **136**, 9608-9618 (2014)), which facilitates the intermolecular charge transport between D1 and A1. Such energy gap controlling strategy have been used in the study of fluorescence in heterodimers (Wang, L. et al. *Nat.Chem.* **9**, 219(2017)). In similar spirit, concerning the resonance contribution from these two vibrational modes, the vibronic coherence is the more plausible candidate. Early observations of quantum beats assigned to electronic coherences due to the beat frequencies observed correspond to the differences in energy between purely electronic exciton states (Thyrhaug, E, et al. *Nat.Chem.* **10**, 780(2018); Ma, F, et al. *Nat.Comm.* **10**, 933 (2019)). However, multiple vibrational modes are also close to the electronic energy differences. Furthermore, electronic decoherence is governed by the overlap of the nuclear wave packet between two states and typically occurs within 10–100 fs. In our

system, the coherence persists in the first 200 fs; this timescale shorter than the electronic decoherence while longer than normally vibrational decoherence time (\sim ps), and this also suggest that the more plausible candidate is vibronic coherence.

Reviewer's summary comment:

I can only recommend this paper for publication in Nature Communications after all these questions have been thoroughly and convincingly addressed.

Response: We greatly appreciate the reviewer's thoughtful review of the manuscript. We have addressed all the comments/concerns raised by the reviewer. Following the reviewer's suggestion, we have substantially modified the manuscript in the discussion section on Page 9-10 and on Page 12-13 of the revised manuscript. In addition, we have polished the language in the revised manuscript. With the additional discussions in addressing the reviewer's comments, as well as the much-modified presentation, the revised manuscript is significantly improved. Hopefully, the reviewer will now find the revised manuscript suitable for publication in Nature Communications.

Reviewer #3 (Remarks to the Author):

The authors have provided an adequate and exhaustive response to the referee's critique. The MS was modified accordingly. Now I believe that the article can be accepted for publication.

Response: We thank the Referee for this assessment.

REVIEWERS' COMMENTS:

Reviewer #1 (Remarks to the Author):

The authors have responded in detail to the questions raised in my report. I now recommend to publish this paper in Nature Communications.